# Sulforaphane: A Broccoli Bioactive Phytocompound with Cancer Preventive Potential

**DOI:** 10.3390/cancers13194796

**Published:** 2021-09-25

**Authors:** Anna E. Kaiser, Mojdeh Baniasadi, Derrek Giansiracusa, Matthew Giansiracusa, Michael Garcia, Zachary Fryda, Tin Lok Wong, Anupam Bishayee

**Affiliations:** Lake Erie College of Osteopathic Medicine, Bradenton, FL 34211, USA; alish56890@med.lecom.edu (A.E.K.); mbaniasadi67397@med.lecom.edu (M.B.); dgiansirac01847@med.lecom.edu (D.G.); mgiansirac27767@med.lecom.edu (M.G.); mgarcia64879@med.lecom.edu (M.G.); ZFryda56890@med.lecom.edu (Z.F.); twong76647@med.lecom.edu (T.L.W.)

**Keywords:** broccoli, isothiocyanates, sulforaphane, cancer, prevention, intervention, molecular mechanisms

## Abstract

**Simple Summary:**

As of the past decade, phytochemicals have become a major target of interest in cancer chemopreventive and chemotherapeutic research. Sulforaphane (SFN) is a metabolite of the phytochemical glucoraphanin, which is found in high abundance in cruciferous vegetables, such as broccoli, watercress, Brussels sprouts, and cabbage. In both distant and recent research, SFN has been shown to have a multitude of anticancer effects, increasing the need for a comprehensive review of the literature. In this review, we critically evaluate SFN as an anticancer agent and its mechanisms of action based on an impressive number of in vitro, in vivo, and clinical studies.

**Abstract:**

There is substantial and promising evidence on the health benefits of consuming broccoli and other cruciferous vegetables. The most important compound in broccoli, glucoraphanin, is metabolized to SFN by the thioglucosidase enzyme myrosinase. SFN is the major mediator of the health benefits that have been recognized for broccoli consumption. SFN represents a phytochemical of high interest as it may be useful in preventing the occurrence and/or mitigating the progression of cancer. Although several prior publications provide an excellent overview of the effect of SFN in cancer, these reports represent narrative reviews that focused mainly on SFN’s source, biosynthesis, and mechanisms of action in modulating specific pathways involved in cancer without a comprehensive review of SFN’s role or value for prevention of various human malignancies. This review evaluates the most recent state of knowledge concerning SFN’s efficacy in preventing or reversing a variety of neoplasms. In this work, we have analyzed published reports based on in vitro, in vivo, and clinical studies to determine SFN’s potential as a chemopreventive agent. Furthermore, we have discussed the current limitations and challenges associated with SFN research and suggested future research directions before broccoli-derived products, especially SFN, can be used for human cancer prevention and intervention.

## 1. Introduction

The development of cancer is a multifactorial process involving cellular mutations, which lead to unrestricted cell growth, thus causing many deleterious effects on the body due to the invasion of malignant cells and metastasis to distant sites, causing widespread organ dysfunction. As a result, cancer is a leading cause of morbidity and mortality across the world [1], which poses a significant burden for our society [2]. Due to the high prevalence of cancer, the utilization of naturally occurring compounds to prevent, inhibit, or reverse tumor development is of high interest in the scientific community. The use of various agents, including natural dietary compounds, is known as cancer chemoprevention, and its major goal is to slow the onset of cancer development and/or to suppress its growth [3]. This brings up an important concept known as “green chemoprevention”, which is defined as the consumption of whole plant foods or their extracts for cancer prevention [4].

A diet high in fruits and vegetables alone can reduce total cancer risk by as much as 14% [5]. Therefore, it is suggested that consuming a well-balanced diet containing a wide variety of vegetables, fruits, whole grains, and other plant-based foods prevents the progression or development of cancer [6,7]. The cancer-preventive potential of dietary agents is believed to be due to the synergy or interactions among bioactive food components or plants’ secondary metabolites [8]. Over 5000 phytochemicals have been isolated from a variety of plants and are identified in vegetables, fruits, whole grains, legumes, and nuts, but most of them remain unknown [8]. Phytochemicals can be divided into specific categories according to their chemical structures, and the most important of these compounds are phenolics and polyphenols, terpenoids, alkaloids, and sulfur-containing compounds. It has been determined that dietary phytochemicals exert cancer-preventive and therapeutic activities through antioxidant, anti-inflammatory, immunomodulatory, antiproliferative, cell cycle-regulatory, cell death-inducing, autophagy-regulating, anti-invasive, antimigratory, and antiangiogenic effects, as well as modulation of various cell signaling pathways [9,10,11,12,13,14,15,16]. Recently, we have provided a broad overview of the recent development of preclinical and clinical research on the cancer-preventive and therapeutic potential of various dietary agents and bioactive food components [17,18,19,20,21,22,23].

A multitude of studies has shown that ingestion of cruciferous vegetables (plants belonging to the Cruciferae family) may lower overall cancer risk, especially for breast, colorectal, bladder, lung, and prostate cancer [24,25,26,27,28]. This is especially true with vegetables in the *Brassica* genus, including broccoli (*Brassica oleracea*), Brussels sprouts, cabbage, cauliflower, and bok choy. Sulfur-containing organic compounds, especially isothiocyanates (ITCs) found within these vegetables, are an important group of phytochemicals that have been shown to have a variety of health benefits. The precursor glucosinolates are metabolized into ITCs by the action of plant myrosinase (EC 3.2.1.147), a β-thioglucoside glucohydrolase, via hydrolysis. ITCs are also released by cutting or chewing, boiling, or by the action of intestinal microflora present in humans [29]. Different glucosinolates produce a variety of distinct ITCs. For example, glucoraphanin (GFN, 4-methylsulfinylbutyl glucosinolate) is the glucosinolate precursor molecule to sulforaphane (SFN, 1-isothiocyanato-4-(methanesulfinyl) butane, Figure 1A). Among the cruciferous vegetables, broccoli and broccoli sprouts have been shown to contain the highest concentration of glucoraphanin, which is also abundant in cabbage, cauliflower, and Brussels sprouts (Figure 1B). Preclinical and clinical studies during the last several decades suggest that ITCs can inhibit carcinogenesis and suppress cancer growth through the regulation of multiple signaling pathways involved in carcinogen biotransformation and detoxification, inflammation, cell cycle, apoptosis, and epigenetic regulation [30,31,32,33,34,35,36].

SFN contains an isothiocyanate functional group (-N=C=S) and a methylsulfonyl side chain (R-(S-O)-R), allowing it to be a water-soluble compound, and its pharmacological activity is increased at the neutral pH of the intestine [37]. In the liver, SFN is metabolized via glutathione (GSH) conjugation to the bioactive compound SFN-N-acetylcysteine that reacts with thiol groups of amino acid residues in a variety of proteins [38]. SFN is known to exert various biological and pharmacological activities, including antioxidant [39], anti-inflammatory, immunomodulatory, and antimicrobial effects [40,41], and is reported to confer various health-promoting and disease-mitigating properties. The beneficial effects of SFN include protection against and/or prevention of gastric ulcer [42], cardiovascular diseases [43], chronic kidney disease [44], aging, and neurodegenerative diseases, including Parkinson’s disease, Alzheimer’s disease, and multiple sclerosis [45,46,47].

Zhang et al. [48] isolated SFN from broccoli, and later its cancer-preventive potential was demonstrated by the same group [49]. Based on studies conducted using cell lines, animal models, and human subjects during the last few decades, SFN is considered a chemopreventive agent with encouraging antineoplastic activities. Although several prior publications provide an excellent overview of the effect of SFN in cancer [50,51,52,53,54,55,56,57,58,59], these reports represent narrative reviews that focused mainly on SFN’s source, biosynthesis, and mechanisms of action in modulating specific pathways involved in cancer without a comprehensive review of SFN’s role or value for prevention of various human malignancies. Several other reviews exclusively focus on a particular cancer type, such as breast [60], bladder [61], or prostate cancer [62]. In addition, more recent review articles [63,64,65] focused on delivery system and synergistic effects of SFN with other anticancer drugs rather than SFN’s individual action as a chemopreventive agent. Therefore, the goal of our systematic study is to present an up-to-date and critical review of SFN’s efficacy in preventing the development or inhibiting the progression of various cancers types with an in-depth analysis of underlying cellular and molecular mechanisms of action. Furthermore, we discuss the current limitations and challenges of utilizing SFN as a dietary compound in humans for cancer prevention and intervention and make suggestions for the future directions of research.

## 2. Bioavailability and Pharmacokinetics of SFN

Phytochemicals are molecules obtained from different kinds of plants that are used for the treatment of diseases in both traditional and modern medicine, and those with poor bioavailability may have limited utility as therapeutic agents [66]. Therefore, understanding the metabolism, pharmacokinetics, and bioavailability of phytochemicals, such as SFN, is vital when considering them as therapeutic agents. SFN-rich powders have been made by drying out broccoli sprout or seed extracts. However, the encapsulation and use of these preparations are extremely expensive and challenging to use in clinical trials due to their instability and required freezing to maintain potency. On a positive note, broccoli and broccoli sprouts possess another major phytochemical of interest, GFN. GFN, a water-soluble glucosinolate and relatively inert precursor of SFN, is contained within broccoli, with the highest amounts present in the seeds and developing florets [67].

GFN is hydrolyzed in vivo to SFN via the myrosinase, which is present in gut bacteria as well as the plant itself [68]. This conversion ranges from 1–40%, with a mean of about 10% [69]. This is attributed to a wide variety of factors that include but are not limited to the mode of delivery, both intra- and inter-individual variations in metabolism and in the microbiome composition and performance of the individuals’ gut, as well as a number of other factors [69]. Studies have been conducted using jejunum from humans in situ, which found that SFN is well absorbed by enterocytes, where it is conjugated with GSH and then secreted back into the lumen [70].

In vivo, SFN is able to be converted to another ITC, erucin, which is a more favorable form in certain tissues [71]. After consumption, SFN is metabolized to cysteine, cysteinylglycine, and finally, N-acetylcysteine (mercapturic acid), all of which are then rapidly excreted in the urine [72]. The excretion process is essentially complete 10–12 h after administration, with maximum concentrations appearing 2–6 h after dosing [72]. Another study reported that SFN is eliminated with small variations between individuals with a typical urinary excretion of 70–90% of the dose [57].

Bricker et al. [73] conducted an experiment where mice were treated with four diets ranging from nonheated broccoli sprout powders (BSP), mildly heated BSP at 60 °C, 5 min steamed BSP, or 3 mmol purified SFN. SFN concentrations in bladder, kidney, skin, lung, liver, and plasma were quantified using high-performance liquid chromatography with tandem mass spectrometry, which showed that mild heating resulted in the greatest ITC metabolite concentrations in vivo followed by the nonheated and steamed BSP diets. They also observed interconversion between SFN and erucin species or their metabolites and reported that erucin is the favored form in the bladder, kidney, and liver, even when only SFN was consumed.

A separate study on rats with oral and intravenous (i.v.) SFN found that the main pathway for metabolic clearance involved conjugation with GSH followed by concurrent processing of the conjugate to mercapturic acid [74]. They measured an absolute bioavailability of 82% between both groups with a dose of 0.5 mg/kg. This study also found that SFN peak plasma concentrations were reached about 1 h after oral administration and within minutes after the i.v. administration. The only time the absorption rate constant decreased was at the highest doses (28 μmol/kg), where the oral bioavailability dropped as low as 20%, indicating that SFN may be absorbed in a carrier-mediated transport mechanism that becomes saturated at high doses. Following i.v. and oral dosing, a rapid drop is observed in the plasma concentrations of SFN, likely reflecting its cellular uptake. This study also shows that the elimination of SFN is illustrated by a long terminal phase, where no major differences in plasma concentrations were apparent between 6 and 24 h following i.v. and oral administrations at lower doses.

Pharmacokinetic studies of SFN have been recorded via cyclocondensation of SFN and its metabolites (dithiocarbamates) with 1,2-benzenedithiol [75]. This highly sensitive method allows for the measurement of SFN and its metabolites in the blood, urine, plasma, and tissues of rodents and humans in the picomolar range [75]. Another study found that after oral administration of 150 μmol SFN to 10-week-old female Sprague Dawley rats, the concentration of dithiocarbamate reached a maximum plasma concentration (*C*_max_) of 60 μM 1 h after dosing, with an elimination half-life of 6.7 h [76].

There have also been several clinical studies elaborating the bioavailability of SFN in humans; however, due to it being difficult to deliver SFN in an enriched and stable form for direct human intake, many researchers use GFN as it is much more reproducible and economical. However, the conversion of GFN to SFN is slow and has high inter-individual variations, with the urinary excretion of SFN typically ranging from 2 to 15% when only GFN is used [69]. Fahey et al. [67] found that co-administering GFN with the enzyme myrosinase in a commercially prepared diet supplement produced an equivalent output of SFN metabolites in the human subjects’ urine to that which was produced when given an equal dose of GFN in a boiled and lyophilized extract of broccoli sprouts [67]. These investigators also found that when the broccoli sprouts or seeds are administered directly to human subjects without prior extraction and inactivation of endogenous myrosinase, SFN in those preparations are 3–4-fold more bioavailable than SFN from GFN delivered without active plant myrosinase. Fahey et al. [69] found that when adding myrosinase to GFN-rich broccoli extracts, the bioavailability of SFN reached levels of about 35–40%. A similar study showed that participants given broccoli sprouts with already-formed ITCs had a larger bioavailability of SFN as well as increased cumulative excretion of its conjugates when compared to broccoli sprout samples with glucosinolates and inactivated myrosinase [77].

Pharmacokinetic studies found that in oral administration of 200 μmol broccoli sprout ITC (SFN) to four healthy human subjects, the *C*_max_ of dithiocarbamate was 1.91 ± 0.24 μM 1 h following the dosing, with a half-life of 1.77 ± 0.13 h, and clearance of 369 ± 53 mL/min [78]. A separate study was conducted involving 20 participants administered 200 μmol SFN as SFN-rich powder in capsules reported a *C*_max_ of 0.7 ± 0.2 µM at 3 h, with a half-life of 1.9 ± 0.4 h for the elimination of SFN equivalents measured by mass spectrometry [79].

In all, SFN is a readily bioavailable and promising agent of interest when considering this phytochemical as a preventative anticancer agent. Although its bioavailability ranges significantly when measured as a metabolite of its precursor, GFN, a co-administration of myrosinase, helps to increase the bioavailability significantly. This elaborates the importance of having myrosinase-producing gut bacteria to aid in uptake and bioavailability when ingested in broccoli. The encouraging results from the pure SFN studies with rats underscore the importance of further human studies regarding the bioavailability of this phytochemical. The results of these studies can then be used to further our knowledge of the best way to utilize SFN as a cancer-preventive agent.

## 3. Toxicity Studies

Xue et al. [80] demonstrated that concentrations of 20–40 μM SFN induced apoptosis and cytotoxicity in human endothelial cells via inhibition of p38 mitogen-activated protein kinase (MAPK), mitogen-activated protein kinase kinase-1 (MAP3K-1), protein phosphatase M3/6, and activation of extracellular signal-related kinase 1/2 (ERK 1/2) and Jun NH_2_-terminal kinase (JNK). Gross-Steinmeyer et al. [81] used SFN on cultured hepatocytes from viable human liver transplants. They determined that exposing human hepatocytes to 10 and 50 μM SFN for 48 h yielded no significant cytotoxicity. Clarke et al. [71] described the effects of 15 μM SFN in normal prostate epithelial cells, which, interestingly, produced no change in expression of p21 and only slight increases of histone deacetylase (HDAC) activity. In this study, normal HDAC activity is an important measurement because it concludes that SFN does not alter the cell cycle in normal, healthy cell lines and therefore is non-cytotoxic. Similarly, p21 increases apoptosis, and hence no change in its expression suggests that SFN does not cause cytotoxicity in normal cells [71]. According to Abbauoui et al. [82], the use of 5, 10, 15, and 20 μM of SFN resulted in no change in survivin expression in normal urothelial cells. Arcidiacono et al. [83] administered 1–5 µg/mL of SFN over 24 h and 48 h periods to normal human epidermal melanocytes and recorded a reduction in cell viability only at the highest concentration (5 μg/mL) used in this study.

An in vivo study conducted by Cornblatt et al. [76] administered a single oral intake of 150 μmol SFN to 10-week-old female Sprague Dawley rats and observed SFN accumulation in mammary tissue with no adverse effects. In mice, dietary SFN at an average daily dose of 7.5 μmol for 21 days had no adverse effects on animal health, food intake, or body weight [84]. This dose was calculated to be the equivalent to the consumption of 1 cup (68 g) of broccoli sprouts in humans [84]. Socała et al. [85] found that at extremely high doses (250–300 mg/kg), SFN caused sedation and muscle impairment in mice. This study concluded that the lethal dose of SFN was 212.67 mg/kg, the therapeutic dose was 191.58 mg/kg, and it also demonstrated that dietary levels of SFN daily showed no changes in animal health, weight, or food intake. Additionally, Castro et al. [86] treated mice with intraperitoneal (i.p.) injection of 50 mg/kg SFN and found no apparent toxicities indicated by lack of change in body weight over 36 days.

Clinical trials on humans have indicated that SFN is relatively safe and free of adverse effects at low doses and minimally harmful at higher doses. However, the Food and Drug Administration limits some clinical trials to 200 μmol of SFN, so more extreme dosages have not been tested [87]. Shapiro et al. [88] showed that consistent dosage in humans of 100 μmol broccoli sprout extract (BSE) or 25 μmol SFN every 8 h for 7 days had no adverse effects measured via thirty-two different parameters in hematological testing. Alumkal et al. [87] determined through a clinical trial that SFN had negligible adverse effects at a dosage of 200 µmol, with the exception of one incident of grade 2 constipation. The study explained that a higher dosage of SFN would likely be of greater benefit but has not been tested yet. Another clinical trial conducted by Tahata et al. [89] found no dose-limiting toxicities of BSE; however, grade 2 nausea occurred in one patient in the 200 μmol SFN dosage group. Zhang et al. [90] conducted a double-blind study with BSE containing 200 μmol SFN and a placebo. Out of the 98 participants, only 3 had adverse effects; two developed headaches and bloating, and the third was in the placebo group. Yagashita et al. [57] found that following oral administration of 100 μmol SFN, patients reported a harsh burning sensation in the back of their throat and posterior aspect of the tongue. At higher doses, patients reported gastrointestinal discomfort, nausea, and heartburn, similar to other clinical trials.

Jeffery and Keck [91] concluded that 3–5 servings of cruciferous vegetables (such as broccoli) per week decreased the risk of developing cancer by over 30%. Even so, many clinical studies have administered SFN in greater concentration than would be found in those 3–5 servings of broccoli and have had success in demonstrating its anticarcinogenic effects with little toxicity. At higher dosages, mild side effects have been reported; therefore, more research on the safety of SFN is warranted. Further research must be performed in order to provide a definitive parameter on safe, maximally effective dosages as well as possible effects of metabolites.

## 4. Sulforaphane in Cancer Prevention and Intervention

### 4.1. Literature Search Methodology

This review evaluates primary research articles that exemplify the anticancer properties of SFN in various cancer models. We have employed the Preferred Reporting Item for Systemic Review and Meta-Analysis (PRISMA) criteria [92] for searching and collecting relevant articles. Primary literature was identified by utilizing PubMed, ScienceDirect, and Scopus, and there were no time restraints on the year of publication. The last search was conducted in April 2021. Various combinations of keywords, such as sulforaphane, broccoli, phytochemicals, cancer, prevention, chemopreventive, tumor, apoptosis, in vitro, in vivo, and clinical studies, were utilized for literature search. Only articles published in the English language were considered for inclusion. Reviews, systemic reviews, meta-analyses, letters to editors, book chapters, conference abstracts, and unpublished observations were excluded. The authors focused specifically on preclinical studies that utilized SFN and excluded reports that used broccoli, kale, watercress, and cauliflower extracts, natural and synthetic analogs of SFN, SFN precursors, and combinations of SFN with other phytochemicals or drugs. However, papers that used SFN in combination with another agent were only included if SFN alone showed statistically significant anticancer properties. Clinical trials utilizing SFN, SFN precursors, and cruciferous vegetable extracts/constituents were searched using clinicaltrials.gov. After reading the abstract to determine relevance, the authors accessed articles through a variety of sources. The full articles were then assessed for in-depth evaluation, and the pertinent information has been summarized and reviewed in the following sections. The overview of literature search and study selection is depicted in Figure 2.

### 4.2. Preclinical Studies (In Vitro and In Vivo)

#### 4.2.1. Breast Cancer

One of the earliest studies to investigate the in vitro cytotoxic effects of SFN on human breast cancer cells was conducted by Tseng et al. [93]. These investigators found inhibition of cell growth when estrogen receptor (ER)-positive and progesterone receptor (PR)-positive MCF-7 cells were exposed to SFN (Table 1). However, involved mechanisms of action were not elucidated. In another study, SFN inhibited proliferation of MCF-7 cells by inducing mitotic arrest in the G2/M phase, increasing cyclin B1 protein and histone H1 phosphorylation, indicating inappropriate cdc2 kinase (CDK1) activation, and inhibiting tubulin polymerization rate [94]. The same researchers [95] uncovered similar mechanisms of inhibited cell growth in F3II sarcomatoid mammary carcinoma cells exposed to SFN. Additionally, Azarenko et al. [96] exposed MCF-7 breast cancer cells to SFN and reported inhibited cell proliferation with a decreased number and size of microtubules at SFN concentrations ≥25 µM.

Ahmed et al. [97] conducted a study to determine the cytotoxic effect of SFN on MDA-MB-231 and MDA-MB-468 breast cancer cells (both are ER- and PR-negative). SFN showed antiproliferative and anti-invasive effects through increased apoptosis; elevated total ubiquitinated proteins (Ub-Prs); inhibition of the activity of the deubiquitinating enzyme (DUBs), ubiquitin-specific protease 14 (USP14), and ubiquitin C-terminal hydrolase L5 (UCHL5); and increased USP14 and UCHL5 proteins. Overall, this study indicated that inhibition of the proteasomal cysteine DUBs activates a feedback reaction that increases the levels of USP14 and UCHL5 proteins and that specific 19S-DUB inhibitors are novel anticancer targets of SFN.

Cao et al. [98] exposed MDA-MB-231, MDA-MB-468, BT-474, and MCF-7 breast cancer cells to SFN and found decreased cell growth via inhibition of the transcription of epigenetic regulator HDAC5 by blockage of the promotor region. This resulted in destabilization of the flavin adenine dinucleotide-dependent histone demethylase 1 (LSD1) protein, indicating that the HDAC5-LSD1 axis is an effective target of SFN in breast cancer cells. Similarly, Royston et al. [99] observed decreased HDACs (HDAC2 and HDAC3) as well as cell cycle arrest in MCF-7 and MDA-MB-231 cells following SFN exposure. This study also showed that SFN decreased histone methyltransferase (HMT) activity in MCF-7 cells, and there was an increase in two tumor suppressors, p53 and p21. Similar cell cycle dysfunctions were noted in MDA-MB-231 and MCF-7 cell lines in a study conducted by Pledgie-Tracy et al. [100], who observed that SFN at concentrations 5 µM and higher inhibited the growth of MDA-MB-231, MCF-7, T47D (ER-positive), and MDA-MB-468 cells. Specifically, MDA-MB-231 and MCF-7 cells were arrested in the G2/M phase in parallel with an increase in cyclin B1 protein expression, and SFN was shown to inhibit global HDAC activity in all cell lines.

Lewinska et al. [101] examined anticancer properties of SFN against MCF-7, MDA-MB-231, and SK-BR-3 (ER-negative, PR-negative, and human growth factor receptor (HER)-positive) cell lines, and found that SFN decreased cell proliferation, which was accompanied by an increase in p21, determined to be p53-independent. Overall, SFN was shown to induce oxidant-based nucleolar stress, which was demonstrated by an increase in superoxide levels, increased protein carbonylation, and changes in nuclear morphology. Lewinska et al. [102] later supported these results by finding elevated levels of p21 in the same three cell lines, as well as increased p53 in MCF-7 cells only. This is the first study to report that SFN-induced cell cycle arrest is permanent, supported by an increase in senescence-associated β-galactosidase staining. Finally, an increase in reactive oxygen species (ROS), genotoxicity, and a decrease in Akt signaling led to apoptosis in all three cell lines.

SFN induced growth inhibition in a time- and concentration-dependent manner in MCF-7 and MDA-MB-231 cells via increased cells in the S and G2/M phases; changes in cell cycle regulatory molecules, such as an increase in p21 and p27; as well as a decrease in cyclin A, cyclin B1, and CDC2 proteins [103]. This study is one of the first to uncover the autophagy-inducing effect of SFN in MDA-MB-231 cells, supported by the formation of autophagosomes, autolysosomes, accumulation of acidic vesicular organelles (AVOs), and an increased level of LC3-II. Later, Pawlik et al. [104] supported these results, showing that SFN induced autophagosomal lysosomes in MCF-7, MDA-MB-231, MDA-MB-468, and SK-BR-3 cells. This effect of SFN has been linked to targets in the pro-survival pathway, indicated by decreases in Akt and S6KI phosphorylation.

Yang et al. [105] demonstrated that 25 µM SFN induced autophagy in three triple-negative (ER-negative, PR-negative, and HER2-negative) breast cancer cell lines, namely, MDA-MB-231, BT549, and MDA-MB-468, as well as suppressed HDAC6 expression, resulting in phosphatase and tensin homolog (PTEN) activation. Another study found decreased expression of cyclin B1, CDC2, p-CDC2, and CDC25C, which may be due to SFN-induced upregulation of the tumor-suppressor gene *Egr1* in various breast cancer cells [106].

Additional cell lines have been exposed to SFN to determine how this phytochemical impacts the cell cycle. Cheng et al. [107] investigated the effects of SFN on ZR-75-1 (ER-positive, PR-positive, and HER2-positive) cell survival and found that SFN decreased cell viability in a concentration-dependent manner. G1/S arrest was observed with concomitant downregulation of CDK2 and CDK4 protein levels at 12.5 and 25 µM SFN. Li et al. [108] transfected normal human mammary epithelial cells to create ER-SH (precancerous) cells and SHR (completely transformed breast cancer) cells. SFN inhibited cell growth in both cell lines, and cell cycle arrest was noted. Additionally, a decrease in HDAC1 was observed, which resulted in an increase in global and local histone acetylation. The ZR-75-1 cell line was used in another study along with MCF-7 cells [109]. Suppression of cell growth was identified in both cell lines after exposure to 30 µM SFN. Additionally, ERα protein expression was significantly inhibited in both cell lines, and Erα mRNA expression and gene transcription were significantly decreased in MCF-7 cells. This was the first study to indicate that regulation of ERα mRNA may be due to SFN inhibition of ERα transcription.

Many other studies have explored the mechanisms behind SFN-induced apoptosis. Pawlik et al. [110] introduced SFN to three ER-positive breast cancer cell lines, namely, T47D, MCF-7, and BT-474, and found decreased cell growth in a concentration-dependent manner, as well as increased PARP cleavage, indicating induced apoptosis. Additional mechanisms that have been attributed to apoptosis include an increase in PARP and caspase-7 cleavage, decreased Bcl-2 protein with increased Bax protein, and an increase in p38 activity with concomitant inhibition of ERK1/2 activity [111]. SFN-induced inhibition of *Bcl-2* was also observed in a study conducted by Hussain et al. [112]. After SFN exposure, they observed decreased viability of MCF-7 cells, and apoptosis was confirmed via observation of morphological changes. Additionally, SFN downregulated the antiapoptotic gene *Bcl-2* and the proinflammatory gene cyclooxygenase-2 (*COX-2**)*. Licznerska et al. [113] exposed MCF-7 and MDA-MB-231 lines to SFN and observed decreased cell viability, induction of apoptosis, and reduced cytochrome P-450 (CYP) 1A1 protein levels. In the MCF-7 cell line, SFN reduced *CYP19* expression and protein levels. However, in MDA-MB-231 cells, SFN increased *CYP19* expression and protein levels, increased CYP1A2 protein levels, and increased aromatase protein. Lubecka-Pietruszewski et al. [114] explored additional proapoptotic mechanisms of SFN in MCF-7 and MDA-MB-231 cell lines. They found elevated *PTEN* and *RARbeta2* expression in both cell lines induced through promotor DNA methylation mechanisms. Additionally, Meeran et al. [115] demonstrated that SFN inhibited proliferation of the same cell lines, which was attributed to decreased human telomerase reverse transcriptase (*hTERT)* via epigenetic modification of the *hTERT* promotor. Finally, Sarkar et al. [116] observed SFN-induced cell growth inhibition and apoptosis in the same cell lines due to decreased expression of heat shock protein 70 (HSP70), HSP90, and heat shock factor 1 (HSF1), increased *p53* and *p21* expression, and increased expression of Bax and Bad with concomitantly decreased expression of Bcl-2.

SFN has also been shown to inhibit cell proliferation by additional cellular mechanisms. Lo and Matthews [117] exposed MCF-7 breast cancer cells to SFN and observed an increase in nuclear factor erythroid-2-related factor 2 (Nrf2), NADPH-dependent oxidoreductase 1 (NQO1), and heme oxygenase 1 (HMOX1) mRNA, indicating that SFN plays an important role in protecting cells from oxidative stress by upregulating phase II detoxifying enzymes. Similarly, Wang et al. [118] exposed the same cell line to SFN and found an increase in thioredoxin reductase 1 (TrxR1) mRNA expression, which plays an important role in protection against oxidative stress. Thangasamy et al. [119] explored the effects of SFN-induced Nrf2 expression on the tyrosine kinase receptor, recepteur d’ origine nantais (RON), also known as macrophage-stimulating 1 receptor, in MDA-MB-231, MDA-MB-468, BT-549, BT-474, SKBR3, and HS578T breast cancer cells. With increased Nrf2 stabilization, RON expression decreased via decreased promoter activity. This was the first evidence depicting SFN-induced decreases in the oncogene RON via Nrf2. In another study, SFN decreased the viability of MCF-7 and MDA-MB-231 cells in parallel with an increase in mRNA and protein expression of the tumor suppressor and oncogene CAV1 [120]. Finally, Castro et al. [86] exposed two triple-negative breast cancer cell lines, MDA-MB-231-Luc-D3H1, and the mouse mammary carcinoma cell line, JygMC(A), to SFN, and found inhibited cell proliferation as well as a decrease in the number of primary, secondary, and tertiary tumorspheres in both cell lines, indicating a reduced capacity for self-renewal.

The anticancer effects of SFN on breast cancer have also been explored in many in vivo studies. Jackson and Singletary [95] subcutaneously injected F3II sarcomatoid mammary carcinoma cells into BALB/c mice. Five days later, lateral tail vein injections of 15 nmol SFN were administered daily for 13 days, after which tumors were excised and examined. The experimenters found significantly smaller tumors in SFN-injected mice versus control mice, as well as reduced proliferating cell nuclear antigen (PCNA) and elevated PARP fragment (Table 2). The BALB/c mice were used in another study, where they were xenografted with MDA-MB-231-luc-D3H1 cells [86]. Daily 50 mg/kg SFN (i.p.) injections were administered for 2 weeks prior to xenograft in one group of mice and for 3 weeks after xenograft in another experimental group. Results showed a 29% decrease in tumor volume in the pretreatment group and a 50% reduction in tumor volume in the posttreatment group when compared to the control. Mechanistic results include decreased expression of *ALDH1A1*, *NANOG*, *CR1*, *GDF3*, *FOXd3*, *NOTCH4*, and *WNT3* genes. Kanematsu et al. [265] transplanted BALB/c mice with KPL-1 cells (ER-positive, PR-negative, and HER2-negative) and injected (i.p.) either 25 or 50 mg/kg SFN 5 days per week for 4 weeks. SFN suppressed the growth of the tumor cells, possibly via induction of apoptosis in a dose-dependent manner. Yang et al. [106] implanted MDA-MB-453 cells into nude mice, and the animals were then treated with 100 mg/kg SFN via i.v. injection daily for 15 days. A significant decrease in tumor weights was observed in the experimental group compared to the control, and an increase in *Egr1* expression was noticed along with a decrease in *cyclinB1* and *CDC25c* expression.

#### 4.2.2. Gastrointestinal Tract and Associate Cancers

##### Esophageal Cancer

SFN has shown anticancer properties in a variety of gastrointestinal tract cancers. Qazi et al. [121] exposed esophageal cancer cell lines OE33 and FLO-1 to SFN and witnessed inhibited cell growth through apoptosis induction, G1 phase arrest, upregulation of p21, and downregulation of HSP90. Additionally, Lu et al. [122] treated EC9706 and ECa109 esophageal squamous cancer cells with SFN and observed inhibited cell proliferation. Increased apoptosis was attributed to activation of the Nrf2 pathway and was accompanied by increases in caspase-9 and LC3B-II, an autophagosome marker, along with decreased p62.

Qazi et al. [121] also extended their in vitro work to evaluate in vivo efficacy of SFN in mice xenografted with BEAC and FLO-1 tumors. After 2 weeks of daily subcutaneous (s.c.) injections of SFN, tumor growth was significantly reduced compared to the control group; however, anticancer mechanisms were not identified. Lu et al. [122] also extended their in vitro findings into a mouse tumor model. BALB/c male mice inoculated with ECa109 cells were given i.p. injections of 5 mg/kg SFN every other day for 2 weeks. Tumor size was decreased in the SFN experimental group, and tissue evaluation revealed an increase in LC3B-II and a decrease in p62. These results, along with the in vitro findings, support the notion that SFN induces apoptosis and promotes autophagy through modulation of the Nrf2 pathway in esophageal cancer cells.

##### Gastric Cancer

The antitumor properties of SFN have also been established in gastric carcinoma. Mondal et al. [123] determined that SFN reduced the viability of AGS gastric carcinoma cells by inducing apoptosis, modifying cell morphology, and generating intracellular ROS. Additional apoptotic mechanisms, including increased Bax, cytochrome c (cyt. c), caspase-3, caspase-8 and PARP cleavage, and decreased Bcl-2, were elucidated. Choi et al. [124] observed proapoptotic mechanisms in the same cell line with an increase in G2/M phase arrest and elevated levels of cyclin B1, p53, p21, phosphorylated AMPK (p-AMPK), intracellular ROS, and cytosolic cyt. c. Dong et al. [125] also witnessed concentration-dependent apoptosis and G2/M phase arrest in both AGS and MGC803 cells after SFN exposure. SFN also inhibited the histone methyltransferase suppressor of variegation, enhancer of zeste, trithorax, and myeloid-nervy-DEAF1 domain containing 2 (SMYD2) and SMYD3 mRNA expression, as well as transcriptional and post-transcriptional regulation of SMYD2 and SMYD3. Finally, Kiani et al. [126] elucidated additional apoptotic mechanisms of SFN in both AGS and MKN45 gastric cancer cell lines. The tumor suppressor genes, caudal type homeobox 1 (*CDX1*) and *CDX2*, showed increased expression at 31 µg/mL SFN; however, *CDX2* expression was significantly reduced at concentrations above 125 µg/mL.

##### Small Intestine Cancer

At least two studies investigated the in vivo anticancer properties of SFN in intestinal neoplasia models using Apc^Min/+^ mice. In one study, the experimental animals consumed 300 and 600 ppm/day SFN via diet over the course of three weeks. Upon tumor assessment, the average number and size of small intestinal polyps in SFN-exposed mice were significantly reduced than the control group in a dose-dependent fashion. Mechanistically, SFN induced apoptosis with a decrease in the expression of p-Akt, p-ERK, and p-JNK [266]. Additionally, Shen et al. [267] fed mice 300 and 500 ppm/day SFN over the course of 10 weeks, and reduced tumor size was mainly contributed to apoptotic mechanisms, including increases in p21, caspase-3, and caspase-9.

##### Colon Cancer

To understand the effect of SFN in colon cancer, many in vitro studies have been conducted using a variety of human colon cancer cell lines. Andělová et al. [127] demonstrated that SFN inhibited the viability and proliferation of SW620 colon cancer cells in a time- and concentration-dependent manner. They found that SFN also caused DNA damage and chromatin condensation after 24 and 48 h and elevated caspase-3 activity with concentrations of SFN above 20 μM. Using SW620 colon cancer cells, Rudolf et al. [128] explored the mechanisms underlying SFN-mediated apoptosis. Results suggested that SFN-induced apoptosis involves DNA-damage signaling with efficiency dependent on p53 status and caspase-2 activation. Enhanced activity of these pathways may serve to amplify the critical proapoptotic signals interacting with mitochondria, which in turn activates effector caspases. Lan et al. [129] examined the effects of SFN in p53-deficient human colon cancer cells SW480 and found that SFN induced mitochondria-associated apoptosis, increased the Bax/Bcl-2 ratio, and activated caspase-3, caspase-7, and caspase-9. Moreover, SFN-induced apoptosis was associated with increased generation of ROS and activation of ERK and p38 MAPK. All in all, SFN-induced apoptosis was confirmed to be ROS-dependent in association with ERK/p38 rather than p53/p73 signaling pathways.

The aforementioned premise has been supported by the work of Gamet-Payrastre et al. [130], who reported that SFN induced cell cycle arrest, followed by apoptosis, which corresponded to an increased expression of cyclins A and B1 in HT-29 colon carcinoma cells. Additionally, the researchers observed no change in the expression of p53 in SFN-treated cells but did observe increased expression of Bax, cytosolic cyt. c, and cleavage of PARP. Comparably, Pappa et al. [131] reported results of increased PARP cleavage in human colon cancer cell lines 40-16 and 379.2 when treated with SFN, and they also observed previously mentioned p53-independent mechanisms of apoptosis induction. Pappa et al. [132] investigated the impact of SFN in 40-16 colon carcinoma cells, and results supported a relationship between SFN and increased PARP cleavage as well as subG1-phase cell-cycle arrest. Additionally, Rudolf and Cervinka [1] extended the role of SFN independent of p53 in human colon cancer HCT-116 cells. By knocking out p53, they observed SFN-dependent cytotoxicity and proapoptotic activity based on selective activation of JNK, which may have directly influenced the expression of Bax and Bcl-2 while promoting the loss of mitochondrial cyt. c and activation of caspases. These results are important in recognizing the chemopreventive potential of SFN in colon cancer with inactivated or lost p53.

In the available literature, SFN has also been shown to have inhibitory effects on the cell cycle in colon cancer cell lines. For example, Byun et al. [134] investigated the effects of SFN on various human colon cancer cell lines, namely, HT-29, HCT-116, KM12, SNU-1040, and DLD-1. Results showed inhibitory growth effects on all cell lines, and SFN was found to induce G2/M phase cell cycle arrest and apoptosis, concomitant with phosphorylation of CDK1 and CDC25B at inhibitory sites, and upregulation of the p38 and JNK pathways. It was also determined that SFN is a potent inhibitor of microtubule polymerization while generating ROS via GSH depletion. On the contrary, Shen et al. [135] demonstrated that SFN inhibited serum-stimulated growth of HT-29 cells by hindering the cell cycle at the G1 phase, in parallel with upregulation of p21^CIP1^ expression and downregulation of cyclin A, cyclin D1, cyclin E, and c-Myc expression.

Zeng et al. [136] concluded that SFN significantly inhibited the proliferation of HCT-116 human colon cancer cells via reduced G1 phase cell distribution and induced apoptosis via enhanced phosphorylation of stress-activated protein kinase (SAPK) and decreased c-Myc. Further expanding upon G2/M phase cell cycle arrest, Pappa et al. [137] demonstrated a novel biphasic inhibitory cell growth pattern in 40-16 human colon cancer cells treated with SFN. A transient SFN exposure for up to 6 h resulted in reversible G2/M cell cycle arrest, while a minimum continuous exposure time of 12 h was necessary for SFN to irreversibly arrest cells in the G2/M phase and subsequently induce apoptosis. These researchers proposed that the reversible G2/M arrest and cytostatic effects of SFN at low concentrations may be related to an observed decrease in GSH induction. In HT-29 human colon cancer cells, Parnaud et al. [138] observed that SFN-treated cells expressed higher levels of p21 and hyperphosphorylation of Rb, leading to increased apoptosis. Moreover, preincubation of HT-29 cells with roscovitine, a cdc2 kinase inhibitor, blocked SFN-induced apoptosis and G2/M arrest, which emphasized the importance of this kinase in the apoptotic pathway induced by SFN.

Many other studies depict the ability of SFN to induce apoptosis in human colon cancer cells. Nishikawa et al. [139] determined that WiDr human colon cancer cells underwent concentration-dependent autophagy as a defense mechanism against SFN-induced apoptosis, as evidenced by the accumulation of acidic vesicular organelles and recruitment of light chain 3 to autophagosomes. Another interesting facet to the proapoptotic effects of SFN was contributed by Chung et al. [140], who investigated the antiproliferation effects of SFN in relation to the oncoprotein SKP2 in various human colon adenocarcinoma cell lines, such as DLD-1, HCT-116, and LoVo. The antiproliferative effect of SFN was accompanied by downregulation of SKP2, leading to the stabilization and thus upregulation of p27KIP1. SFN treatment also led to the activation of both Akt and ERK, indicating downregulation of SKP2 without Akt or ERK inhibition.

In addition to the proapoptotic effects of SFN summarized in previous sections, other investigators have proposed additional possible anticancer mechanisms of SFN in colon cancer cells. Traka et al. [141] published a transcriptome analysis of human colon Caco-2 cancer cells exposed to physiological concentrations of SFN, recording a >2-fold increase in expression of 106 genes and a >2-fold decrease in expression of 63 genes, supporting the role of SFN in inhibiting cell growth. Most notably, upregulation in several genes associated with antioxidant response element (ARE)-mediated transcription and Nrf2 activation, including NQO1, theoredoxin reductase (TR1), aldo-ketoreductase (AKR), and heme oxygenase 1, was observed. Remarkable genes that experienced significant decreases in expression post-SFN exposure included formyltetrahydrofolate synthase and DNA (cytosine-5-)-methyltransferase 1. Another transcriptomic study conducted by Johnson et al. [142], using SFN-treated human colon cancer cell lines, HT-29 and HCT-116, demonstrated that SFN strongly induced the expression of NQO1, several other Nrf2-dependent targets, and Loc344887 (NMRAL2P), a noncoding RNA that acts as a novel, functional pseudogene for the NmrA-like redox sensor and coregulator of NQO1.

Many other studies have identified additional anticancer mechanisms of SFN that further the understanding of this biochemical. In the investigation of the role of SFN on autophagy, Wang et al. [143] specified that SFN induced autophagy in a concentration- and time-dependent manner in Caco-2 cells. Specifically, the anticancer effects of SFN on Caco-2 cells may be attributed, at least in part, to induction of various phase II enzymes, namely, uridine 5′-diphospho-glucuronosyltrasferase 1A1 (UGT1A1), UGT1A8, and UGT1A10, via nuclear translocation of Nrf2 and human pregnane X receptor (hPXR). Additionally, Harris and Jeffery [144] investigated the effects of SFN on the expression of multidrug resistance protein 1 (MRP1) and MRP2 in the same cell line. SFN at 5 µM significantly increased the expression of MRP2 but showed no effect on the expression of MRP1. The relatively small concentration of SFN required to produce these effects is a significant distinction to understand when considering the possible physiological consequences of consuming SFN in relation to preventing the occurrence of colon cancer.

Continuous efforts to explain the chemoprotective effects of SFN have led researchers to examine its effects in association with other biochemical pathways that may contribute to its anticancer potential. Using human colon cancer cell line HCT-116, Rajendran et al. [145] demonstrated that SFN inhibited HDAC activity and increased HDAC protein turnover, causing susceptibility to SFN-induced DNA damage. As a result, the researchers again offered a model for the differential effects in cancer cells versus non-cancer cells of HDAC inhibition and DNA damage/repair signaling following SFN treatment. Martin et al. [146] supported these results by observing decreased HDAC and hTERT mRNA levels in RKO and HCT-116 cells after SFN exposure. Okonkwo et al. [147] investigated SFN and its structural analogs as modifiers of HDAC and histone acetyltransferase activity (HAT), as well as anticancer effects on human colon cancer cell lines HCT-116 and SW480. In SFN-treated HCT-116 cells, an increase of nuclear pH2AX and pRPA32 levels were observed, suggesting both enhanced DNA damage and repair, respectively. In the SW480 colon cancer cell line, SFN demonstrated increased cytotoxicity when compared to normal CCD112 colon epithelial cells. Additionally, increases in p300, a HAT-associated protein, expression and histone H4 acetylation were observed in both cell lines treated with SFN.

A study conducted by Bessler and Djaldetti [148] addressed the effects of SFN on the inflammatory relationship between human peripheral blood mononuclear cells (PBMCs) and the colon cancer cell lines, HT-29 and RKO. Results showed that while HT-29 and RKO cancer cells stimulated both pro- and anti-inflammatory cytokine production by PBMCs, the addition of SFN exerted a concentration-dependent inhibitory effect on inflammatory cytokine production by these cells, specifically with tumor necrosis factor-α (TNF-α), interleukin-1β (IL-1β), IL-6, IL-2, IL-10, and interferon γc (IFNγc). Tafakh et al. [149] investigated the expression of many genes at the mRNA level in HT-29 cells treated with SFN. Results indicated that SFN preconditioning decreased the expression of COX-2, microsomal prostaglandin E synthase-1 (mPGES-1), hypoxia-inducible factor (HIF-1), vascular endothelial growth factor (VEGF), C-X-C chemokine receptor type 4 (CXCR4), matrix metalloproteinase-2 (MMP-2), and MMP-9. Additional novel findings include decreased PGE_2_ generation and inhibited in vitro motility/wound healing activity of HT-29 cells. The researchers concluded that the anticancer effects of SFN were associated with antiproliferative and antimigratory activities arising from the downregulation of the COX-2/mPGES-1 axis.

Although the exact anticancer mechanisms of SFN remain to be fully clarified, additional studies have served to progress the current agenda of understanding the effects of SFN on colon cancer cells. Jakubíková et al. [150] found that at high concentrations, SFN induced accumulation of sub-G1 cells, cell death, and dissipation of mitochondrial membrane potential in Caco-2 cells. Mechanistically, phosphorylation of ERK1/2 and Akt kinases was increased, but SFN had no effect on JNK and p28 activation. These results highlight the importance of phosphatidylinositol 3-kinase/protein kinase B (PI3K/Akt) and mitogen-activated protein kinase/ERK kinase (MEK)/ERK signaling as intracellular mediators in SFN-mediated phase II enzyme transcription and cell cycle arrest in Caco-2 cells. Expanding upon the apoptotic effects of SFN, Kim et al. [151] explored how the oxidation of sulfur in the side chain of SFN affected apoptosis induction in human colon cancer cell lines, HCT-116, LoVo, Caco-2, and HT-29. Researchers found that SFN, which contains oxidized sulfur, elicited a greater growth inhibitory effect in comparison with SFN analogs containing non-oxidized sulfur. The data demonstrated that increased apoptosis induction in HCT-116 cells by SFN was associated with an increase in caspase-8 activation but not with a rise in caspase-9 activity.

Additionally, Kim et al. [152] investigated the relationship between SFN and HIF-1α expression in HCT-116 human colon cancer cells, and results showed a concentration-dependent inhibition of HIF-1α expression and suppression of HIF-1α target gene activation. SFN also inhibited VEGF expression, suggesting that SFN may hinder colon cancer progression and angiogenesis by downregulating the expression of HIF-1α and VEGF.

The vast majority of publications regarding the effects of SFN colon cancer are based on in vitro studies; however, a handful of in vivo studies exist. Expanding on their in vitro findings, Byun et al. [134] reported that SFN markedly suppressed the growth of HCT-116 xenografted tumors in nude male mice. Mechanistically, SFN increased cyclin-dependent kinase 1 (CDK1), MAPK-activated protein kinase 2 (MK2), and p38 phosphorylation. Myzak et al. [245] also expanded their in vitro work by treating APC^min/+^ mice with a single dose of SFN (10 µmol) and reported suppressed tumor growth as well as significant inhibition of HDAC activity with an associated increase in acetylated histones H3 and H4. Additionally, long-term treatment with SFN for 10 weeks in the diet resulted in elevated levels of acetylated histones and p21^WAF1^ in the colon, including acetylated histones specific to the promoter region of *P21* and *Bax* genes. These results suggest that HDAC inhibition by SFN contributes to the chemopreventive and chemotherapeutic mechanisms of SFN in vivo. Similarly, Rajendran et al. [268] investigated the antitumor capability of SFN to induce Nrf2-dependent pathways and inhibit HDAC activity in vivo. By treating wild type (WT) and Nrf2-deficient (Nrft2^−/+^) mice with the colon carcinogen dimethylhydrazine (DMH) and subsequent dietary SFN (400 ppm) treatment, researchers demonstrated that WT mice were more susceptible to colon tumor induction than Nrf2^−/+^ mice. WT mice also had higher levels of HDAC on several genes, including cyclin-dependent kinase inhibitor 2a (Cdkn2a/p16). These results ultimately support the role of SFN in inducing Nrf2 pathways in colon cancer models and should lead future studies to focus on Nrf2 as a tumor growth determinant and HDAC inhibitor.

##### Hepatocellular Cancer

A study conducted by Yu et al. [153] using HepG2 human hepatic cancer and Hepa1c1c7 hepatoma cells found that SFN treatment exhibited cytotoxicity via an increase in the expression of the MAPK/ERK2 signaling pathway. A similar study using HepG2 cell lines showed that SFN treatment demonstrated antiproliferative effects by upregulating metallothionein (MT) genes MT-1 and MT-II and promoting apoptosis, indicated by induction of caspase-3, Bax, and PARP cleavage, as well as decreased levels of Bcl-2 and Bcl-xL expression [154]. Park et al. [155] observed very similar mechanisms within the same cell line. Keum et al. [156] elaborated different antiproliferative mechanisms of SFN in the same cell line and determined that increased ARE caused an increase in heme oxygenase-1 (HO-1) through activation of Nrf2 and suppression of Kelch-like ECH-associated protein 1 (Keap1).

Jeon et al. [157] treated Huh-7, SNU-449, and NCTC hepatic cancer cell lines with SFN and found an increase in apoptosis via induction of caspase-3, caspase-8, and caspase-9. They were also able to show several novel findings, including increased G2/M phase arrest, as well as decreased expression of phosphofructo-2-kinase/fructose-2,6-biphosphatases (PFKFB) and HIF-1α, leading to decreased VEGF and angiogenesis. Liu et al. [158] also showed that SFN treatment in HepG2 resulted in decreased expression of HIF-1α and VEGF, along with decreased expression of signal transducer and activator of transcription 3 (STAT3).

Moon et al. [159] reported treatment of Hep3B hepatic cancer cells with SFN elicited an increase in apoptosis via the production of ROS. They also observed several unique findings, including a decrease in telomerase and hTERT, which may be related to an inhibition of Akt signaling. Wu et al. [160] demonstrated similar results in HepG2 cells as well as the unique findings that SFN treatment decreased expression of Vimentin and increased expression of E-cadherin, suggesting that SFN suppresses epithelial–mesenchymal transition (EMT). Moreover, SFN treatment in HepG2 cells resulted in apoptosis via regulation of several novel pathways, including increased expression of Bip/glucose-regulated protein 78 (GRP78), XBP-1, caspase-12, C/EBP-homologous protein (CHOP)/GADD153, and Bid [161]. These findings, along with the other studies, imply SFN’s role in inducing apoptosis in hepatocellular cell lines.

A separate group of researchers investigated SFN’s role in activating phase I biotransformation enzymes in Hepa 1c1c7 (murine) and HepG2 cells and found that it reduced cell viability by increasing the expression of CYP1A1 mRNA [162]. They also determined that SFN successfully activated AhR transformation and its subsequent binding to the xenobiotic response element (XRE).

Very limited information is available on in vivo anti-hepatocellular cancer effects of SFN. However, a study was conducted on female BALB/c athymic mice xenografted with HepG2 cells and treated with SFN for 13 days, and the investigators noted a subsequent reduction in tumor growth and volume via an unknown mechanism [160].

##### Pancreatic Cancer

An early study found that pancreatic cancer cell lines MIA PaCa-2 and PANC-1 treated with SFN resulted in a promotion of cell cycle arrest and death. These effects were mediated via increased apoptosis with subsequent induction of pro-apoptotic proteins (caspase-3 and caspase-8), G2-M arrest, and ROS [163]. A similar study conducted with the aforementioned cell lines along with AsPC-1 and BxPc-3 cells found cell cycle arrest via suppressed Akt signaling and CDK4 expression as well as a rise in the proteasomal degradation of HSP90 client proteins [164]. Another group of researchers found that SFN treatment of AsPC-1, BxPc-3, Capan-1, and MIA PaCa-2 cells reduced overall cell viability by increasing apoptosis, but also by decreasing nuclear factor-κB (NF-κB) binding [165]. Rodova et al. [166] found similar results of increased apoptosis in ASPC, PANC-1, and human pancreatic cancer stem cells (CSCs) through various mechanisms, including increased caspase-3, decreased Bcl-2, and decreased Nanog, Oct4, Smo, and Hedgehog (Hh) signaling.

Yin et al. [167] treated AsPC-1, BxPc-3, PANC-1, and MIA PaCa-2 cell lines with SFN and demonstrated increased apoptosis through a novel mechanism of increased miR-365a-3p. Moreover, elevated expression of miR135b-5p was identified in a separate study, which ultimately led to the upregulation of the RASAL2 tumor suppressor gene involved in the inhibition of tumor growth [168]. Chen and colleagues [169] noted inhibition of cellular proliferation, invasion, and migration of PANC-1 and MIA PaCa-2 cells treated with SFN. This was demonstrated via an increase in apoptosis, AMPK signaling, ROS production, E-Cadherin, HO-1 and Nrf-2, and a decrease in Vimentin and N-cadherin.

Forster et al. [170] reported a reduction in cell viability in BxPc-3 and AsPC-1 pancreatic cell lines, which was contributed to increased E-Cadherin, Cx43, gap junctional intercellular communication (GJIC), as well as decreased expression of cancer stem cell markers c-Met and CD133. These mechanisms aligned with those reported by Georgikou et al. [171], who noted increased Cx43 and GJIC, increased expression of the GJA1 mRNA gene whicthath encodes Cx43, and decreased miR30a-3p in PANC-1 cells.

Several in vivo studies have been conducted to elaborate on SFN’s antiproliferative effect against various pancreatic cancer cell lines. Kuroiwa et al. [269] reported that SFN prevented N-nitrosobis(2-oxopropyl) amine (BOP)-induced pancreatic carcinogenesis in male Syrian hamsters via an unknown mechanism. Pham et al. [163] exposed male severe combined immunodeficiency (SCID) mice inoculated with PANC-1 cells to SFN over the course of 3 weeks and reported decreased tumor growth. Another study treated female athymic (nu/nu) mice inoculated with MIA PaCa-2 with SFN and reported inhibition of tumor growth [164]. Reduced tumor growth was also reported in a study utilizing male NOD/SCID/IL2Rγ mice inoculated with human pancreatic CSCs when treated with SFN for 6 weeks [270]. Mechanistically, they noted an upregulation in E-Cadherin, as well as a decrease in VEGF, Bcl-2, Smo, Gli1/2, and PDGFα [270]. Kallifatidis et al. [165] conducted a study exposing nude mice inoculated with MIA PaCa-2 cells to SFN and reported inhibition in tumor growth and angiogenesis and increased apoptosis concomitant with decreased NK-κB expression. Chen et al. [169] treated BALB/c nude mice with SFN and reported a reduction in tumor volume and weight via activation of AMPK signaling. They also reported a strengthening of Nrf-2 nuclear localization, implying SFN’s potential for cancer prevention and treatment.

#### 4.2.3. Gynecological Cancers

##### Cervical Cancer

All in vitro studies examining the effect of SFN on cervical cancer were performed using HeLa cells. Park et al. [156] exposed HeLa cells to SFN and observed inhibited cell viability, increased formation of apoptotic bodies, increased accumulation of cells in the sub-G1 phase, and downregulated Bcl-1/Bcl-xL and c-inhibitor of the apoptosis (cIAP-1). Another study reported that SFN resulted in 50% inhibition of HeLa cell growth following SFN treatment at 12 μM. Mechanistically, SFN induced a concentration-dependent increase in caspase-3 and a downregulation of Bcl-2, COX-2, and IL-1β [172]. A subsequent study by Khan et al. [173] found that SFN inhibited DNA methyltransferase (DMNT), downregulated DNMT3B, and decreased HDAC activity by directly interacting with HDAC1 in HeLa cells. This study concluded that SFN has potential antitumorigenic effects and may reactivate silencing of tumor suppressor genes epigenetically by altering methylation. Moreover, Cheng et al. [174] determined that SFN treatment of HeLa cells decreased their survival and proliferation by inducing cell arrest in the G2/M and G1 phases as well as decreasing cyclin B1 and cyclin B1/CDC2 complexes.

##### Endometrial Cancer

Recently, Rai et al. [175] investigated the effects of SFN on MFE280, KLE, Ishikawa, Hec1B, Hec1A, MFE296, and AN3CA endometrial cells. Inhibition of proliferation and induction of apoptosis was observed along with G2/M phase arrest, which was attributed to both the suppression of the PI3K/Akt/mTOR pathway and increased phosphorylation of MEK and ERK, leading to decreased MEK and ERK expression. This was the first study to imply that SFN has anticancer properties against endometrial cancer.

Rai et al. [175] also expanded their investigation to an in vivo Ishikawa xenograft mouse model and observed decreased tumor volume via apoptotic mechanisms after i.p. administration of 50 mg/kg SFN daily for 16 days. The antitumor effect was superior to that of paclitaxel (10 mg/kg, once every seven days).

##### Ovarian Cancer

One of the earliest studies to investigate the in vitro cytotoxic effect of SFN on ovarian cancer was conducted by Chaudhuri et al. [176]. The findings revealed the antiproliferative effect of SFN on SKOV3, C3, and T3 cells with decreased cell survival and loss of cell viability with increased concentration. Mechanistically, SFN had an impact on the PI3K/Akt pathway by downregulating the steady-state level of the total and active Akt protein, cyclin D1, cdk4, and cdk6 levels in all three cell lines. In another study, exposure of OVCAR-3 and SKOV-3 cells to SFN for a two-day period significantly reduced cell viability and the accumulation of cells in the G1 phase, and cell apoptosis was identified after 4 h of SFN treatment [177]. Bryant et al. [178] exposed SKOV3 and MDAH-2774 cells to SFN, which resulted in a concentration-dependent inhibition of cancer cell proliferation. Additionally, SFN induced apoptosis and increased S phase cells and G1 arrest in MDAH-2774 cells. Kim et al. [179] reported the effect of SFN on cell growth at 72 h in OVCAR3, OVCAR4, OVCAR5, and SKOV3 cell lines and found that SFN was effective at inhibiting cancer cell growth via activation of p38 and ERK. In a separate study, SFN treatment over a 24 h period induced apoptosis in SKOV3 and A2780 ovarian cancer cell lines. Further investigation uncovered an increase in type 1 inositol 1,4,5- triphosphate receptor (IP_3_R1) and Nrf2 protein expression with a resultant increase in Nrf2-regulated genes, such as the catalytic subunit of glutamate-cysteine ligase (GCLC), HMOX1, and NQO-1. Additional findings in the A2780 cell line include increased ROS and increased phosphorylation of HSP27, JNK, mitogen-activated protein kinase 1 (MEK1), p38, p90RSK, and c-JUN [180]. Finally, Chang et al. [181] reported that SFN treatment decreased cell survival and proliferation, increased cell accumulation in the G2/M phase, and downregulated cell division cycle protein 2 (CDC2), causing dissociation of the cyclin B1/CDC2 complex in PA-1 ovarian cancer cells.

The only study investigating the in vivo anticancer effect of SFN on ovarian cancer was conducted by Hudecova et al. [180]. Athymic nude mice with A2780 ovarian cancer xenografts received 40 mg/kg SFN (i.p.) for 7 days, after which tumor size was significantly reduced compared to the control. These results were attributed to increased IP_3_R1, supporting the in vitro results as mentioned earlier.

#### 4.2.4. Hematological Cancers

##### Leukemia

Fimognari et al. [182] performed one of the earliest studies to investigate the in vitro effect of SFN on leukemia in which SFN arrested Jurkat cells in the G2/M phase. SFN induced apoptosis in a time- and concentration-dependent manner with the appearance of decreased DNA content as well as increased expression of p53 and Bax proteins. In a subsequent study, Fimognari et al. [183] supported these findings with very similar results in the same cell line. In addition, cyclin D3 was significantly decreased, while the expression of cyclin D2, CDK4, and CDK6 was slightly decreased. A further study by Fimognari et al. [184] reported that SFN induced differentiation in HL-60 promyelocytic leukemia cells, including granulocytes and macrophages, via apoptosis. Choi et al. [185] reported that SFN inhibited the viability of U937 cells, induced apoptosis, caused accumulation of cells in the sub-G1 phase, downregulated Bcl-2, increased expression of caspase-3, stimulated the release of ROS, and hyperpolarized the mitochondrial membrane in a concentration-dependent manner. Koolivand et al. [186] reported that treatment of U937, KG-1, HL-60, and NB-4 acute myeloid leukemia cells with SFN decreased live cells and increased mortality rates in a concentration- and time-dependent manner. In addition, increased concentrations of SFN induced primary apoptosis of HL-60 cells, and SFN significantly decreased the expression of miR-155 in all four cell lines. Moreover, a study by Shang et al. [187] revealed the effect of SFN on HL-60 cells and reported antiproliferative effects previously reported, but also noted increased Fas-associated death domain (FADD), which is indicative of apoptosis. A similar study that focused on the effect of SFN on HL-60 cells reported that SFN induced NQO1 expression in cells containing NQO1*2 genotype and SFN downregulated cytosolic Keap1, accompanied by increased nuclear Nrf2. Prolonged exposure of HL-60 to SFN resulted in inhibition of cell proliferation due to cell cycle arrest and apoptosis caused by induction of p-H2AX expression, PARP cleavage, and caspase-mediated cell death [188]. Suppipat et al. [189] utilized different leukemia cell lines, namely, Nalm-6, Nalm-6 human pre-B cells, and REGM and RS-4 pre-B ALL cells, and observed decreased cell viability and accumulation of cells in the G2/M cell cycle phase after exposure to SFN. A study by Prata et al. [190] revealed that SFN has cytotoxic effects at 30 μM in leukemia B1647 cells and significantly decreased cell viability. Moreover, 10 μM of SFN significantly decreased aquaporin-8 (AQP8) both at transcriptional and protein levels, decreased intracellular ROS levels, and lowered the level of Nox-2. Misiewicz et al. [191] investigated the effect of SFN on the L-1210 leukemia cell line, and the results revealed cell growth arrest, decreased cell viability, and decreased cell density.

##### Lymphoma

Only one study investigating the in vitro effect of SFN on lymphoma was conducted by Ishiura et al. [192]. In this study, SFN significantly inhibited the proliferation of Kaposi’s sarcoma-associated herpesvirus-positive primary effusion lymphoma cell lines (BC2, BC3, and HBL6) and decreased their viability by suppressing p38 MAPK and Akt signaling. In addition, SFN caused the cleavage of caspase-3, caspase-7, caspase-9, and PARP in BC2 and BC3 cells, which indicates that SFN triggers apoptosis via the caspase 9-dependent pathway.

#### 4.2.5. Lung Cancer

A preliminary study conducted by Mi and Chung [193] examined the effect of SFN on the A549 human lung cancer cell line, and they found increased apoptosis and cell cycle arrest at the G1/S and G2/M checkpoints. Furthermore, SFN inhibited cell division by causing a decrease in tubulin polymerization; therefore, preventing mitotic spindle formation during mitosis. Utilizing the LTEP-A2 human lung adenocarcinoma cell line, similar results were obtained, as SFN significantly inhibited cellular proliferation by upregulating apoptosis and halting the cell cycle at the G2/M phase [194]. Zuryn et al. [195] corroborated these results in the A549 cell line and determined that part of the cell cycle arrest was mediated by a decrease in cyclin D1 and upregulation of the CDK inhibitor p21. In a later study, this same group revealed that SFN caused cell cycle arrest by decreasing cyclin B1 and increasing cyclin K in H1299 cells [196]. Similar results were uncovered when A549 and H1299, non-small cell lung cancer cells (NSCLCs), were exposed to SFN, and it was determined that an increase in apoptosis was partially mediated by increased p53 and Bax [197]. Additionally, SFN exposure caused significantly increased H3 and H4 acetylation and decreased HDAC activity. Gao et al. [198] further elucidated SFN’s ability to alter the epigenetics of A549 cells, in which SFN showed downregulated HDAC1, HDAC3, HDAC6, decreased CpG methylation, and upregulated histone modifier H3K4me1.

Zhu et al. [199] further elucidated SFN’s mechanisms of regulating gene expression, apoptosis, and cellular growth. In A549 and H1299 cell lines, SFN decreased miR-19a and miR-19b and inhibited the transcription regulators Nanog and Oct4. This group determined that apoptosis was upregulated by increased Bax, caspase-3, caspase-8, and caspase-9. Additionally, cellular growth was restricted by the downregulation of the Wnt/β-catenin pathway. Wang et al. [200] observed similar results utilizing the NSCLC cell lines H1299, 95C, and 95D. Tsai et al. [201] determined that A549 and CL1-5 cells treated with SFN showed significantly inhibited growth by downregulation of multiple growth pathways, including β-catenin, Akt, and FAK. In addition to the mechanisms listed previously regarding apoptosis, this team determined that there was increased externalization of phosphatidylserine to the outer lipid bilayer. In another study, SFN increased the proapoptotic factor p53, upregulated modulator of apoptosis (PUMA), Bax, caspase-9, and p73, and simultaneously decreased the antiapoptotic factor Bcl-2 in XWLC-05 cells [202]. Similar antiapoptotic mechanisms were observed in cadmium-transformed BEAS-2BR, bronchial epithelial cells, treated with SFN [203]. Moreover, this group of researchers was the first to determine SFN’s ability to induce autophagy and decrease the transcription factor Nrf2 in bronchogenic carcinoma cell lines.

Chen et al. [204] was the first group of researchers to determine SFN’s ability to induce proteasomal activity in various NSCLCs, such as PC9/gef, H1975, A549, CL1-5, and H3255. Furthermore, novel mechanisms involving the inhibition of cellular growth were confirmed by observing SFN’s ability to significantly decrease the phosphorylation of the epidermal growth factor receptor (EGFR) and STAT3 signal transducer in addition to decreasing p-Akt as previously established. These investigators illustrated for the first time that ERK5 activation via phosphorylation mediates SFN’s suppressive effect on human bronchogenic carcinoma cells. Additionally, SFN exhibited significant potential to inhibit cellular adhesion and migration via downregulation of pc-jun, pc-Fos, Snail1, MMP-2, and N-cadherin in spite of SFN inducing expression of tight junction, ZO-1, and cellular adhesion molecule E-cadherin [205]. Geng et al. [206] further confirmed these results by treating SK-1 and A549 cells to SFN and reported an increase in 26S proteasomal activity in addition to upregulation of the ERK1/2 signaling cascade. As previously described, these investigators demonstrated that SFN stimulated apoptosis via an increase in Bax and caspase-3 while decreasing the antiapoptotic factor Bim.

Recently, it has been reported that SFN hinders the acquisition of tobacco-smoke-induced lung cancer stem-cell-like properties via modulation of the IL-6/ΔNp63α/Notch signaling axis [207]. This axis is upregulated in tobacco-smoke-induced lung cancer cell lines HBE and A549, and an increase in ΔNp63α is positively correlated with CD133 and Oct4 expression. SFN administered over a period of 7 days was shown to significantly downregulate ΔNp63α, resulting in a concomitant decrease in CD133 and Oct4. Furthermore, SFN’s ability to attenuate the Notch signaling pathway was exemplified by its capability to decrease IL-6, NICD, Hes1, and Nanog expression.

One of the earliest studies to determine SFN’s antitumorigenic effects in vivo was performed on A/J mice treated with lung carcinogen benzopyrene and 4-(methylnitrosamino)-1-(3-pyridyl)-1-butanone (NNK). Benzopyrene- and NNK-induced mice were treated with 1.5 and 5 µmol/g SFN (per os, p.o.) for 42 weeks, resulting in a significantly reduced tumor size and weight. Mechanistically, apoptosis was upregulated via increased caspase-3 and downregulated PCNA [271].

Liang et al. [194] utilized nude mice inoculated with LTEP-A2 human lung adenocarcinoma cells to investigate the antitumor effects of SFN. Parenteral administration (i.p.) of 25, 50, and 100 mg/kg SFN for 9 days resulted in a reduced tumor weight via upregulation of apoptosis and increased cell cycle arrest at the G2/M restriction point. Jiang et al. [196] further elucidated SFN’s ability to decrease tumor weight in NOD/SCID mice inoculated with A549 human lung cancer cells. SFN (9 µmol/day, p.o.) for 28 days caused an increase in proapoptotic factors p53, p21, and Bax. Furthermore, this team of researchers determined that SFN’s capacity to alter chromosomal packaging in vivo was mediated by increased H3 and H4 acetylation and downregulation of HDAC activity. BALB/c nu/nu male mice inoculated with NSCLCs and treated with SFN at a dose of 10 µmol/kg, five times per week, over 21 days, showed significantly reduced tumor volume [204]. This decrease in tumor volume was mediated by decreasing the tumor’s response to growth factors via downregulation of the EGFR. In a later study, this same group determined novel mechanisms of SFN’s action in vivo using BALB/c nude female mice inoculated with H1299 cells [205]. ERK5 growth signaling pathway was significantly inhibited, thus resulting in a decreased tumor weight and volume. Additionally, cell migration and invasion were hindered by the downregulation of N-cadherin adhesion protein despite the upregulation of E-cadherin and tight junction ZO-1, as previously reported in vitro. Wang et al. [200] corroborated these results utilizing nude male BALB/c mice inoculated with H1299 and 95D cells. Mice treated with SFN every 3 days for 4 weeks at a dose of 25 or 50 mg/kg (i.v.) showed a reduced number of metastatic lung nodules via downregulation of N-cadherin, Vimentin, and β-catenin.

#### 4.2.6. Neural Cancer

There have been only a handful of studies investigating the effect of SFN treatment on neural cell cancers, and the results show promise for anticancer properties. Karmakar et al. [208] performed a study treating glioblastoma cell lines T98G and U87MG with SFN and reported a decrease in cell viability and increase in apoptosis via upregulation of caplain, proapoptogenic mitochondrial protein Smac/Diablo, apoptosis inducible factor (AIF), as well as increased expression of caspase-3, caspase-9, caspase-12, Bax:Bcl-2, cyt. c, and increased intracellular calcium level. This group of researchers also found a reduction in the expression of apoptosis inhibitor proteins and NF-κB. Another study conducted by Bijangi-Visheshsaraei et al. [209] exposed U87, U373, U118, and SF767 cells to SFN and found a decrease in cell survival and promotion of cell death via induction of caspase-3, caspase-9, and caspase-7, and increased production of ROS with a resultant increase in double-stranded DNA breaks. A novel finding in their study showed increased expression of γ-H2AX, a protein that localizes near DNA strand breaks and recruits other proteins to the site of damage. Miao et al. [210] were able to show that treatment with SFN in U87 and U251 glioblastoma cells resulted in inhibition of cell survival and promotion of cell death via many of the above mechanisms. Zhang et al. [211] exposed U251MG glioblastoma cells to SFN and observed a reduction in cell viability and promotion in cell death via increased apoptosis and decreased invasion linked to decreased expression of MMP-2, MMP-9, and Galectin-3. In a separate study, Li et al. [212] treated U87MG and U373MG cells with SFN and documented reduced proliferation, migration, and invasion via decreased expression of MMP-2, as well as an increase in ERK1/2.

Only one study has reported the anticancer potential of SFN using an in vivo glioblastoma model. Bijangi-Visheshsaraei et al. [209] treated female NSG mice inoculated with GBM10 cells with 100 mg/kg SFN for 5-day cycles (p.o.) over 3 weeks and found increased inhibition of tumor growth; however, mechanisms were not revealed.

#### 4.2.7. Skin Cancer

In one of the earliest studies investigating the effects of SFN on skin cancer, Misiewicz et al. [191] treated ME-18 human melanoma cells with SFN and reported arrested cell growth. This was accompanied by increased apoptosis with resultant DNA strand breaks and phosphatidylserine externalization. Arcidiacono et al. [83] also showed suppressed cell growth as well as decreased invasion and metastasis in 501MEL human malignant melanoma cells via increased expression of caspase-3, caspase-8, and caspase-9. They also reported that SFN upregulated the expression of pro-apoptotic genes, including p53, Bax, PUMA, Fas, MDM2, EGR1, GADD45b, ATF3, and CDKN1A. Subsequently, the same research team observed that SFN treatment shifted the growth factor receptor ratio from prosurvival to proapoptotic one, with a measurable increase in apoptosis in A375 human malignant melanoma cells [213].

In a separate study, Mantso et al. [214] demonstrated that when treated with SFN, A375 cells exhibited decreased survival due to an increase in apoptosis with concomitant expression of multiple caspases. Another experiment conducted by Fisher et al. [215] on A375 and WM793 cell lines showed that SFN reduced spheroid migration, formation, and invasion in addition to increased apoptosis via suppression of Ezh2, H3K27me3, Bmi-1, and Suz12.

Hamsa et al. [216] treated B16F-10 melanoma cell lines with SFN and witnessed a reduction in cell viability and increased apoptosis via induction of caspase-3, caspase-8, caspase-9, and Bax, as well as decreased expression in Bcl-2, IL-1β, IL-6, TNF-α, IL-12p40 C-Fos, ATF-2, CREB, and NF-κB. Enriquez et al. [217] also reported a reduction in cell viability when treating B16 murine melanoma cells with SFN with a measurable decrease in HDAC. Decreased HDAC activity was reported in another study conducted on B16 and S91 murine melanoma cells [218]. Additionally, SFN decreased cellular proliferation, promoted oxidative stress, and modulated gene expression in Bowes and SK-MEL-28 human melanoma cell lines [219]. Mechanistically, an increase in apoptosis was observed through upregulation in phosphorylated p38 kinase, p53, PUMA, Bax, and ROS production.

Several in vivo studies elaborate on SFN’s in vitro anticancer effects. Thejass and Kuttan [272] treated C57Bl/6 mice with xenografted B16F-10 melanoma tumors with 500 µg/kg (i.p.) SFN over 10 days and reported inhibition of tumor growth and lung metastasis via decreased expression of hydroxyproline, uronic acid, and hexosamine in the lungs as well as decreased sialic acid and γ-glutamyl transpeptidase (GGT) in the serum. Ancillary studies indicate that SFN treatment inhibited the spread of metastatic tumor cells via stimulation of cell-mediated immune response, upregulation of IL-2 and interferon-γ (IFN-γ), and downregulation of several proinflammatory cytokines [273]. Another study was conducted on C57BL/6 mice inoculated with B16 murine melanoma cells and treated with 500 µmol/kg SFN 3 doses/week (i.p.) for 4 weeks and reported a reduction in tumor volume via decreased expression of HDACs [218]. An additional study using the same tumor model and SFN regimen also reported inhibition of tumor growth and reduced cell volume via a decrease in HDAC expression [217]. Fisher et al. [215] conducted an experiment using NSG mice inoculated with A375 melanoma cells and treated with SFN for 6 weeks and noted a decrease in tumor formation and volume. A mechanistic evaluation showed increased apoptosis via decreased MMP-9, MMP-2, H3K27me3, and Ezh2, as well as elevation in the cleavage of procaspase-8, procaspase-9, and PARP.

#### 4.2.8. Urogenital Cancers

##### Bladder Cancer

Shan et al. [220] was the first group to portray the anticancer properties of SFN in a bladder cancer cell line. SFN inhibited T24 cell growth by inducing early apoptosis, decreasing cells in both the S and G2/M phases, and reducing p27 expression. In a separate study, this group demonstrated SFN’s ability to downregulate COX-2 mRNA and protein levels in the T24 cell line, which was shown to be a consequence of increased nuclear NF-κB translocation with a concurrent decrease in COX-2 promotor binding as well as upregulated phosphorylation of p38 [221]. Shan et al. [222] continued this research by establishing SFN’s capability of inducing thioredoxin reductase-1 (TR-1) and glutathione S-transferase (GSTA1-1) via activation of p38 MAPK. Furthermore, SFN has been shown to significantly inhibit T24 cell adhesion to matrigel, fibronectin, and laminin and attenuate cell migration [223]. Mechanistically, SFN decreased COX-2, MMP-2, and MMP-9 while upregulating E-cadherin. Suppression of Snail and ZEB1 transcription factors, mediated by increased miR-200c expression, has been linked to the increase in E-cadherin [222].

The T24 bladder carcinoma cell line has been exposed to SFN in another study conducted by Jo et al. [224]. SFN inhibited cell viability and induced apoptosis by activating caspase-3 and caspase-9, increasing PARP cleavage, upregulating Bax, and increasing cytosolic cyt. c levels. Additionally, SFN increased endoplasmic reticulum stress, evident by increased GRP78 and CHOP. SFN-induced apoptosis was also initiated in RT4, J82, and UMUC3 bladder cancer cell lines in a study conducted by Abbaoui et al. [82]. Moreover, the same research group displayed SFN’s capability to inhibit HDAC1, HDAC2, HDAC4, and HDAC6 activity; however, histone acetylation status was not significantly altered [225]. These results suggest that SFN-induced HDAC inhibition may not directly impact histone acetylation but instead may act on additional cytoplasmic targets.

SFN-induced apoptosis has been explored in two additional bladder cancer cell lines. BIU87 cells exposed to concentrations greater than 20 µM SFN demonstrated significant inhibition of cell proliferation, which was attributed to apoptosis indicated by an accumulation of cells in the G2/M phase of the cell cycle [226]. Additionally, this study illustrates that SFN-enhanced insulin-like growth factor-binding protein-3 (IGFBP-3) also plays a role in inducing apoptosis in bladder carcinoma. Park et al. [227] documented SFN’s capability of inducing apoptosis by disrupting the mitochondrial membrane and mediating intracellular ROS. After the 5637 bladder cancer cell line was exposed to SFN, an increase in G2/M phase arrest was observed along with increased H3 phosphorylation, PARP cleavage, cyclin B1, Cdk1, caspase-3, and a decrease in mitochondrial membrane potential.

SFN has been shown to inhibit bladder tumor growth by inducing apoptosis in two in vivo studies. Abbaoui et al. [82] inoculated nude mice with UMUC3 bladder carcinoma cells, and after 2 weeks of dietary SFN (295 µmol/kg) exposure, there was a significant decrease in tumor weight compared to the control. This study also confirmed detectable SFN levels in the mouse plasma and tumor tissue; however, mechanisms of tumor suppression were not explored. Wang and Shan [274] observed similar antitumorigenic effects in UMUC3 xenografted mice fed 12 mg/kg SFN per day. SFN induced apoptosis, promoted the expression of caspase-3 and cyt. c, and suppressed expression of survivin compared to the control group. This is one of the earliest reports to suggest that survivin is a target of SFN in bladder carcinoma in vivo.

##### Prostate Cancer

An early study executed by Brooks et al. [228] determined that SFN inhibited cellular proliferation in various human prostate cancer cells by increasing the antioxidant enzymes NQO1, quinone reductase, and GST. Hahm et al. [229] found that SFN reduced viability of DU145, LNCaP, PC-3, and CWR22v1 cells by suppressing the IL-6/JAK/STAT3 pathway and decreasing Bcl-2, which, in turn, enabled apoptosis and decreased cell growth. Choi et al. [230] demonstrated that SFN reduced cellular growth in LNCaP and PC-3 cells by increasing proapoptotic factors Bax and apoptotic protease activating factor-1 (APAF-1) and decreasing antiapoptotic factors BAK, Bcl-xL, cIAP1/2, and XIAP. Additionally, cell cycle regulators p53 and E2F1 were increased. Similarly, Carrasco-Pozo et al. [231] discovered increased apoptosis in LNCaP cells after SFN exposure via decreased BcL-xL, prostate-specific antigen (PSA), nuclear androgen receptor, and HIF-1α. Kim and Singh [232] similarly reported a decrease in LNCap and C4-2 cellular proliferation after SFN exposure due to an increase in transcriptional repression of androgen receptor mRNA, leading to a decrease in Ser210/213 phosphorylated and total androgen receptor.

A study performed by Herman-Antosiewicz et al. [233] found that SFN induced autophagy and apoptosis in PC-3 and LNCaP cells by upregulation of LC-3 cleavage, which stabilizes the autophagosome. Additionally, apoptosis was increased due to an increment in cyt. c release. Xiao et al. [234] also observed LNCaP and PC-3 cell growth inhibition via increased apoptosis with a concomitant increase in Bax, cyt. c, ROS, and LC-3 cleavage. Watson et al. [235] observed similar mechanisms of SFN-induced autophagy.

SFN increased apoptosis and G2/M phase arrest in DU145 human prostate cancer cells by increasing JNK pathway activity, leading to activation of p53 and increased JNK, ROS, and PARP cleavage [236]. Hac et al. [237] also conducted a study exposing PC-3 cells to SFN and observed apoptosis and S-phase arrest most likely caused by an increase in DNA double-stranded breaks. Another study conducted by Lee et al. [238] also documented decreased LNCaP cell viability and growth through an increase in PARP cleavage and caspase-3 after SFN exposure, indicating increased apoptosis. Singh et al. [239,240] showed that SFN reduced PC-3 and DU145 cell survival and proliferation, which was marked by increased caspase-3, caspase-8, caspase-9, and Bax, as well as decreased Bcl-2 with the promotion in PARP cleavage. Negrette-Guzmán et al. [241] determined that SFN resulted in reduced PC-3 cell viability, measured by decreases in PGC1α and HIF-1α along with increases in Bax and NRF1.

Ahmed et al. [97] reported a novel finding that SFN inhibited the growth of 22Rv1 cells by inducing apoptosis through increased USP14 and UCHL5 proteins. Beaver et al. [242] determined that SFN-induced cell growth restriction in LNCaP and PC-3 cells was related to a decrease in CDK2, PLK1, and BMX, which are all required for cell cycle progression. Additionally, SP1 was downregulated, and NQO1 was upregulated. Similarly, a study conducted by Bhamre et al. [243] showed reduced growth of LNCaP cells coinciding with increased G2/M cell cycle arrest and subsequent apoptosis through decreased levels of Jun protein. Additional findings indicated increased levels of NQO1, TXNRD1, GSTM1, MGST1, SOD1, and PRDX1.

A study conducted by Clarke et al. [71] found that SFN inhibited the growth of LNCap and PC-3 cells through downregulation of HDAC3, HDAC4, and HDAC6. All these mechanisms contributed to the inhibition of the cell cycle at the G2/M phase, leading to decreased cell growth and division. Similarly, Gibbs et al. [244] found that SFN decreased the viability of LNCaP and VCaP cells via a decrease in HDAC6, androgen receptor expression, HSP90, and ERG. Another study similarly found decreased levels of HDAC and increased levels of acetylated H3 and H4 leading to G2/M phase arrest [245]. Zhang et al. [246] reported similar results regarding the decreased expression of HDAC and elevated levels of acetylated H3 but also found upregulation of Nrf2 and NQO-1, as well as decreased DNMT3a and DNMT1.

Beaver et al. [247] observed decreased cellular proliferation of LNCaP and PC-3 cells upon SFN treatment, and mechanistically, they found altered expressions of approximately 100 long non-coding RNAs. Hahm et al. [248] exposed LNCaP and PC-3 cells to SFN and also found reduced cell proliferation and migration concomitant with decreased Notch1, Notch2, and Notch4 as well as an increase in DNA fragmentation. After SFN exposure, Hsu et al. [249] observed decreased cellular proliferation of the LNCaP and PC-3 cell lines, and they also saw a decrease in DNMT1 and DNMT3. The investigators also found a decrease in cyclin D2 promoter methylation and consequently an increase in cyclin D2, associated with the prevention of tumor progression. According to Wong et al. [250], similar mechanisms were found to be responsible for SFN’s cytotoxicity in the LNCaP and PC-3 cell lines in spite of an increase in CCR4 and transforming growth factor-β (TGF-β) receptor type 1 (TGFBR1), which allow for tumor progression. Vyas et al. [251] also observed decreased cellular proliferation and viability in the same cell lines after SFN exposure due to decreased CD24, ITGA6, ZEB2, and c-Myc. A study conducted by Watson et al. [252] found decreased cell viability in LNCaP and PC-3 cells introduced to SFN as well as an increase in post-translational modification of SUV39H1. This subsequently led to a decrease in H3K9me3-specific histone methyltransferase, resulting in an increase in CD8 + T-cells. Singh et al. [253] reported that SFN resulted in decreased cell viability through decreased glycolysis in LNCaP and 22Rv1 cells. Moreover, there was a decrease in LDHA, which facilitates the glycolytic process by converting pyruvate to lactate and is an important biomarker for cancer progression. Finally, PMK2, a pyruvate kinase important in the glycolysis pathway, was also decreased.

Several studies showed a reduction in prostate cancer cell migration through different and novel mechanisms. Peng et al. [254] revealed that treatment of DU145 cells with SFN resulted in decreased pseudopodia, as well as decreased MMP-2 and CD44v6. This study also showed increased p-ERK1/2 and E-Cadherin. Vyas and Singh [255] determined that SFN decreased cell proliferation and migration of PC-3 and DU145 cell lines with a concomitant reduction in E-Cadherin, as well as an increase in PAI-1 and vimentin.

Hac et al. [256] exposed the PC-3 cell line to SFN and revealed a decrease in S6K1 phosphorylation, which is important for the regulation of mTOR and protein synthesis. LC3, a key protein in autophagy, was shown to be elevated as well. Pei et al. [257] showed an increase in H2S (associated with DNA damage), p38 (involved in cell differentiation, apoptosis, and autophagy), and JNK (implicated in tumor suppression) upon SFN treatment of PC-3 cells. Wiczk et al. [258] conducted a study using SFN and PC-3 cells, which revealed increased S6K1 dephosphorylation leading to decreased levels of mTOR and survivin protein. Two studies conducted by Xu et al. [259,260] revealed that SFN reduced PC-3 cell viability through different mechanisms. The first study reported decreased NF-κB levels, p65 nuclear translocation, VEGF, and IKKα and IKKβ phosphorylation [259]. The second study [260] showed an increase in p-ERK1/2, JNK1/2, and p-c-Jun, which led to reduced cell viability. However, increased AP-1 was also recorded, which is critical in allowing cancer migration and proliferation. ELK1, a factor that is coupled to androgen receptors allowing growth of prostate cancers, was also increased. Yao et al. [261] found that SFN increased JNK and ERK signaling and decreased VEGF and HIF-1α, all contributing to a decrease in proliferation of DU145 cells.

A study conducted by Singh et al. [262] showed inhibited LNCaP and 22Rv1 cellular proliferation through a variety of mechanisms. The effects of SFN on these cells showed a reduction in carnitine palmitoyltransferase 1A, medium-chain acyl-CoA dehydrogenase, hydroxyacyl-CoA dehydrogenase trifunctional multienzyme complex subunit α, and sterol regulatory element-binding protein 1. Herman-Antosiewicz et al. [263] conducted a study that showed that SFN exposure to LNCaP cells promoted cell cycle arrest and decreased proliferation. Mechanistic results revealed decreased levels of cyclin D1, cyclin E1, CDK4, CDK6, CDK1, and CDC25C, as well as increased levels of p53, p21, and cyclin B1. Abbas et al. [264] found that SFN induced cell cycle arrest, mediated via several mechanisms. Decreased hTERT was reported, and there was also a decrease in HDAC-inhibitory activity, allowing histones to be deacetylated to reduce DNA transcription. This study also measured decreased H3K4me2 and MeCP2. This reduction in cellular proliferation was observed even with an increase in H3K18A, a biomarker in cancer progression, as well as increases in DNMT3a, DNMT1, pan-acetylated H3, and H4.

The promising in vitro anticancer effects of SFN have been extended to several in vivo investigations. Myzak and colleagues [84] conducted a study that revealed a reduction in tumor growth in nude, male, athymic BALB/c mice that were xenografted with PC-3 cells and treated with 443 mg/kg/day SFN for 3 weeks. Increases in Bax and acetylated-H3 and -H4 were identified alongside decreased HDAC. Similarly, Singh et al. [240] found inhibited tumor growth in male and female athymic mice with xenografted PC-3 tumors using 5.6 µmol SFN 3 times/week (p.o.) for 3 weeks. Mechanistically, increased apoptosis via increased Bax was identified. Subsequently, Singh et al. [275] found a decrease in tumor cell proliferation and pulmonary metastasis in male TRAMP F1 hybrid mice using 6 µmol SFN in 0.1 PBS 3 times a week (p.o.) for 17–19 weeks. The study also reported increased apoptosis and increases in Bax, Bid, Bak, Bad, and PARP cleavage. Antiapoptotic factor Mcl-1 was decreased, and E-cadherin was also upregulated [275]. Traka et al. [276] revealed inhibition of cell viability and proliferation using 0.1 or 1 µmol/g SFN per day (p.o.) for 8 weeks in PTEN^ L/L and PB-Cre4 mice. Elevated levels of caspase-3, caspase-7, and cyclin B1 were reported, as well as decreased cyclin D2.

Singh et al. [262] treated adenocarcinoma of TRAMP mice SFN, which resulted in tumor growth inhibition. Decreased acetyl CoA carboxylase (ACC1) and FASN, enzymes involved in fatty acid metabolism, were reported. Additionally, fatty acid, acetyl-CoA, and phospholipid levels were decreased. Similarly, Singh et al. [253] introduced 1 mg SFN 3 times per week (p.o.) for 5 weeks to TRAMP and Hi-Myc mice with prostate adenocarcinoma and noted inhibition of tumor growth. The TRAMP mice model overall showed a lower incidence of prostatic intraepithelial neoplasia by 23–28%. A mechanistic evaluation revealed a decrease in hexokinase 2, the rate-limiting enzyme in glycolysis, a decrease in tumor M2-pyruvate kinase, a pyruvate kinase specific for malignant growths, and a decrease in lactate dehydrogenase, an enzyme responsible for anaerobic glycolysis in both mouse models. Additionally, there was significantly suppressed glycolysis in the Hi-Myc mouse model.

### 4.3. Clinical Studies

Many retrospective (observational), prospective, and interventional clinical studies have been conducted to evaluate cancer-preventive and therapeutic efficacy of broccoli and broccoli-derived products containing GFN and/or SFN. In the following section, we present many of these studies.

Epidemiological studies have suggested that consumption of cooked meat and meat products increases the risk of colorectal cancer [277]. Walters et al. [278] investigated the cancer-preventative capability of cruciferous vegetables by examining excretion of the food-derived carcinogen, 2-amino-1-methyl-6-phenylimidazo[4,5-*b*]pyridine (PhIP), in a group consisting of twenty non-smoking Caucasian males. During phases 1 and 3, participants avoided cruciferous vegetable intake, and during phase 2, participants ingested 250 g each of Brussels sprouts and broccoli per day. At the end of each phase, each participant consumed a meat meal containing 4.9 μg PhIP, and the urinary metabolite of PhIP (*N*^2^-hydroxy-PhIP-*N*^2^-glucuronide) was measured. There was a significant increase in urinary excretion *N*^2^-hydroxy-PhIP-*N*^2^-glucuronide in phase 2 compared to phases 1 and 3 (Table 3). This study demonstrates that consumption of cruciferous vegetables can induce the metabolism of PhIP in humans, underscoring chemopreventive potential.

Due to consumption of foods contaminated with aflatoxin, a dietary hepatocarcinogen, and exposure to high levels of phenanthrene, a polycyclic aromatic hydrocarbon, the residents of Qidong, People’s Republic of China, are at an elevated risk for the development of hepatocellular carcinoma. In a randomized, placebo-controlled chemopreventive trial, 200 healthy adults from Qidong drank hot water infusions of 3-day old broccoli sprouts containing 400 μmol of GFN for two weeks. The results indicated an inverse association between the urinary excretion of dithiocarbamates (SFN metabolites) and aflatoxin-DNA adducts or *trans*, *anti*-phenanthrene tetraol (a metabolite of phenanthrene) in individuals receiving the broccoli drink [279]. In a crossover clinical study conducted by the same research group [280], 50 healthy volunteers from the same region as mentioned above received two broccoli-sprout-derived beverages (enriched with SFN or GFN). It was determined that individuals receiving either one or both beverages had a 20–25% increase in excretion of GSH-derived conjugates of benzene, acrolein, crotonaldehyde (all airborne pollutants) compared with their preinterventional base values, indicating enhanced detoxification of environmental carcinogens. In an extended study by the same group [281] with a larger study population and extended period of intervention, the broccoli sprout beverage elicited rapid and sustained increases in the levels of excretion of GSH conjugates of benzene and acrolein, but not crotonaldehyde, compared to the placebo group [281]. These findings may be valuable in designing a future chemopreventive trial to evaluate the ability of broccoli bioactive food components to prevent environmental carcinogenesis.

A crossover clinical study by Riso et al. [282] evaluated the protective effect of 200 g broccoli intake for 10 days in healthy males, including 10 smokers and 10 non-smokers. Blood samples were collected at 0, 10, 30, and 40 days for assessment of DNA damage, IGF-I, and HDAC. The results showed that the broccoli diet decreased DNA strand breaks in both groups and reduced the oxidized purines only in the smoker group. However, broccoli intake did not alter HDAC activity or IGF-I levels. In the following year, the same group [283] published the results of another study in which 27 healthy smokers consumed steamed broccoli (250 g/day) or a control diet for 10 days. Broccoli intake decreased the level of oxidized DNA lesions and prevented hydrogen peroxide-induced DNA strand breaks in PBMCs from smokers. A higher level of protection was observed in participants with the *GSTM1*-null genotype.

Several epidemiological studies indicate that diets rich in cruciferous vegetables are linked to a reduced risk of oral cancer [293,294]. In a pilot clinical trial in 10 healthy volunteers, consumption of either GFN- or SFN-rich BSE or topical exposure to SFN-rich extract showed upregulation of *NQO1* mRNA in buccal scrapings (oral mucosa), suggesting the chemopreventive potential of SFN against carcinogen-induced oral cancer [284].

In a study conducted by Atwell et al. [79], 20 healthy adults consumed fresh broccoli sprouts or myrosinase-treated BSE, each providing 200 μmol SFN single dose daily, or two 100-μmol doses taken 12 h apart in a divided dose phase. Three chemopreventive mechanistic targets of SFN, namely, HO-1, HDAC, and p21, were measured in the PBMCs. The results indicated that the consumers of both sprouts and BSE had fluctuations in the HDAC activity, with greater decreases in HDAC activity noted in higher-dose or repeated-intake subjects.

In a double-blind, randomized controlled trial, 54 women with abnormal mammograms who were scheduled for breast biopsy consumed 250 mg of broccoli seed extract daily containing 30 mg of GFN (68.57 μmol GFN) or a placebo for 2–8 weeks before their biopsy. Plasma and urine SFN metabolites, PBMC HDAC activity, and tissue biomarkers, such as HDAC3, HDAC6, H3K18ac, H3K9ac, Ki-67, and p21, were measured pre- and post-treatment in benign, ductal carcinoma in situ (DCIS) or invasive ductal carcinoma (IDC) breast tissues. Statistically significant reductions in PBMC HDAC activity, as well as a decrease in Ki-67 and HDAC3 serum levels in benign tissues of the GFN-supplemented group, were observed, indicating that GFN may modulate HDAC activity, resulting in decreased cell proliferation [285]. The same group [286] conducted another study on 54 women with abnormal mammograms who were scheduled for breast biopsy and investigated the relationship between the intake of cruciferous vegetables and selected tumor biomarkers of the previous study. Total cruciferous vegetable intake was assessed using the National Cancer Institute Diet History Questionnaire and Arizona Cruciferous Vegetable Food Frequency Questionnaire, urine and serum ITC levels were estimated, and the biomarkers were measured in breast tissues using immunohistochemistry. Participant cruciferous vegetable intake was 81.7 g/day, and increased total cruciferous vegetable intake was associated with significantly lower Ki-67 levels in breast DCIS tissues but not in benign or IDC tissues. These results suggest that consumption of cruciferous vegetables plays a role in inhibiting breast cancer cell proliferation. A phase II clinical trial is investigating the effect of α-cyclodextrin complex of SFN known as SFX-01 (Evgen Pharma, Alderley Park, United Kingdom) on 60 participants, given over the 18 months. The preliminary results indicate that SFX-01 has the potential to reverse resistance to endocrine therapies in patients with ER+ HER2- metastatic breast cancer [287].

A prospective randomized, double-blind clinical trial is investigating the effect of freeze-dried broccoli sprouts containing 90 mg SFN (507.64 μmol SFN) daily on 40 patients with non-resectable pancreatic ductal adenocarcinoma undergoing palliative chemotherapy for a year. The primary goal of the study is to increase the overall survival of the patients [288]. The final results are not published yet.

Epidemiological studies have suggested that individuals who consume diets rich in cruciferous vegetables are at lower risk of both the incidence and aggressive forms of prostate cancer [295,296]. A clinical trial conducted by Traka et al. [289] investigated the effect of the consumption of 400 g broccoli per week for 12 months on prostate gene expression in 20 males with high-grade prostatic intraepithelial neoplasia. The results indicated that there was a significant difference in gene expression between *GSTM1*-positive and null individuals in the broccoli diet group. The altered genes are associated with the TGF-β1 and epidermal growth factor (EGF) signaling pathway. The findings suggested that consuming broccoli interacts with the GSTM1 genotype through signaling pathways associated with inflammation and carcinogenesis in the prostate. Later, Alumkal et al. [87] performed a study in which a BSE containing 200 µmole SFN was given to 20 men with recurrent prostate cancer for 20 weeks. The findings demonstrated that treatment with SFN-rich extract reduced PSA by more than 50% only in one patient, and a smaller decline in PSA was noted in seven patients. Additionally, there was a significant increase in PSA doubling time (PSADT) due to the intervention. Cipolla et al. [290] performed a double-blinded, randomized, placebo-controlled multicenter trial in which 75 men with increasing PSA levels after radical prostatectomy received 60 mg (338.42 μmol) of oral SFN daily for 6 months followed by 2 months with no treatment. The results showed that the SFN group had lower PSA levels at months 0, 1, 3, and 6, as well as longer PSADT compared to the placebo group. In addition, a PSA increase of more than 20% was noted in the placebo group at 6 months compared to the SFN group. Traka et al. [291] demonstrated the effect of weekly intake of 300 mL portion of soup made from GFN-enriched broccoli or standard broccoli (control) for 12 months on prostate cancer patients who underwent transperineal template biopsy. Gene expression in tissues from the patients was quantified before and after the dietary intervention. The results indicated that the consumption of GFN-rich broccoli soup affected gene expression in the prostate of patients, indicating a reduction in the risk of cancer progression. In another study conducted by Zhang et al. [90], 96 men scheduled for prostate biopsy consumed capsules containing BSE (providing 200 µmol SFN) daily or a placebo with the objective of understanding the impact of SFN on blood HDAC activity, prostate genes expression, and tissue biomarkers, such as histone H3 lysine 18 acetylation (H3K18ac), HDAC3, HDAC6, Ki67, and p21. The results showed a significant increase in urinary and plasma SFN metabolite levels in the extract-treated group but failed to show any significant difference in HDAC activity. However, within the subgroup of subjects with confirmed prostate cancer, the extract treatment significantly increased HDAC activity. In addition, there was no significant difference between the two groups on tissue biomarkers, which was thought to be due to undetectable levels of SFN in prostate tissue. Finally, using biopsy samples, this study detected treatment-related alterations in the expression of six genes, including α-methylacyl-coA racemase (AMACR), androgen receptor-regulated long non-coding RNA (ARLNC1), C1orf64, SLIT1, RP11-672G23-1, and RP1-274L7.1, which may play a role in the development of prostate cancer and are important targets for future studies.

A double-blinded clinical study by Kirkwood et al. [292] is investigating the effect of BSE on 17 individuals for 28 days on patients with a history of melanoma and multiple atypical/dysplastic nevi. Patients randomly received oral BSE (standardized for 50, 100, or 200 μmol SFN) daily for evaluation of its effect on melanoma risk marker STAT3 as well as proliferative marker Ki-67 and apoptotic marker Bcl-2. The final results are not published yet. However, according to a report published by the same group [89], the aforementioned oral regimen of BSE has been implicated in decreasing pro-inflammatory cytokines human interferon-inducible protein 10 (IP-10), monocyte chemoattractant protein 1 (MCP-1), monokine induced by gamma (MIG), and macrophage inflammatory protein 1β (MIP-1β), along with decreased IFN-γ and increased tumor suppressor decorin in skin nevi of patients with a history of melanoma. Additionally, a trend towards decreased nevi size was observed with increased SFN level after the 28-day period; however, measurements were not statistically significant [89].

Although not many clinical studies have investigated the effect of cruciferous vegetables and cruciferous vegetable products on cancer, the research that has been conducted shows promising results, such as decreased cell proliferation, impacts on inflammatory cytokines, and decreases in tumor markers. These studies have opened up varied directions for future research that can more accurately identify the benefit of intake of cruciferous vegetable components, especially SFN.

## 5. Conclusions and Future Directions

The *Brassica* genus includes broccoli, Brussels sprouts, and cabbage, among other nutritious vegetables that contain organosulfur compounds, including ITCs. SFN has been shown to be the most potent ITC and has been found in the highest concentrations in broccoli and broccoli sprouts. SFN possesses anticancer properties that have lately been a focus of natural product cancer preventative research. The purpose of this review was to provide a systemic analysis of preclinical and clinical studies that examined the anticancer and chemopreventive actions of SFN. This analysis also detailed the bioavailability and toxicity of SFN.

While bioavailability studies are lacking due to SFN instability and the expense of creating SFN products, a substantial amount of in vitro, in vivo, and clinical trials utilizing SFN precursors and metabolites have supported the use of this phytochemical as an anticancer agent. After oral consumption, SFN has been detected in serum and many tissue types indicating its ability to reach a variety of tumors. In vitro and in vivo studies have aided in determining a therapeutic dose of SFN; however, there has been a disconnect in the literature between dosages utilized in animal models and the tolerable doses in humans. Many of the published in vivo studies report chemopreventive effects with doses of SFN that would be unachievable in human subjects. Doses ranging from 5 to 100 mg/kg SFN have been shown to suppress tumor growth in animal models, translating to 350–7000 mg for a 70 kg individual. Additionally, one of the most common delivery methods employed in animal studies was i.p. injection, but most clinical trials aim to investigate oral intake. Future research in animal studies should aim to translate better into clinical trial scenarios to provide more insight into the true anticancer effects of SFN.

One of the most pressing questions in this research is the translation of cruciferous vegetable intake to GFN/SFN dose consumption for cancer prevention. Yagishita et al. [57] have reported an average concentration of 0.38 μmol/gram GFN in raw broccoli from Baltimore supermarkets with a range of 0.005–1.13 μmol/g. Additionally, an average of 0.36 μmol/g GFN has been reported in field/greenhouse-grown broccoli [57]. Most clinical trials utilize doses of GFN ranging from 25 to 800 μmol [57], translating to about 65–2105 g raw broccoli or 3/4 to 23 cups of raw broccoli. The lower end of this range is reasonable to consume daily, but the mid-upper end of this range breaches a realistic boundary, opening the opportunity for GFN/SFN supplements that meet the required chemopreventive doses.

There have been few adverse effects of SFN reported in the literature. One study has reported a lethal dose of SFN determined via oral SFN administration to rats, but such toxic doses have not been established in clinical trials due to maximal dose regulations. Mild adverse effects, such as gastrointestinal distress, nausea, and heartburn, have been reported in clinical trials. However, the reported anticancer properties of SFN greatly outweigh these minute side effects.

It was determined that the vast majority (i.e., 192) of the current literature focusing on anticancer properties of SFN has been preclinical studies, whereas only 19 studies have investigated the effects of products containing SFN and its biogenic precursor in clinical trials. In vitro studies have focused on numerous specific cancer subtypes, while in vivo studies have used only a few organ-specific tumor models. Prostate cancer has been the most investigated cancer type, closely followed by breast cancer, based on published clinical trials. Analysis of the published data has revealed SFN’s diverse mechanistic regulation, which includes, but is not limited to, modulation of proliferation, cell death, cell cycle arrest, oxidative stress, inflammation, migration, invasion, and metastasis (Figure 3).

In regard to in vitro studies, a few mechanistic pathways of note were consistent between cancer subtypes. It is clear that the Nrf2/Keap pathway plays a significant role in SFN induced cytotoxicity. Mechanistically, SFN has been shown to increase nuclear Nrf2 through two specific pathways. The first is through modification of Keap1 cysteine residues, causing a release of Nrf2 that is able to translocate to the nucleus. The second is through epigenetic modulation of HDACs and DNMTs. SFN has been shown to influence epigenetic modulation through DNA methylation, histone modification, and non-coding RNAs, allowing for increased Nrf2 transcription and translation, again resulting in increased nuclear Nrf2. Within the nucleus, Nrf2 is able to bind to ARE and maf to increase transcription of cytoprotective phase II enzymes (Figure 4). Additionally, apoptosis was induced by SFN in the majority of in vitro studies. Many mechanisms were uncovered, including, but not limited to, increased Bax and Bad, increased caspase-3, caspase-7, caspase-8, caspase-9, and decreased Bcl-2 and Bcl-xL (Figure 5). Modulation of these endpoints indicates that SFN acts on both the intrinsic and extrinsic apoptotic pathways. SFN was also recognized to induce cell cycle arrest in many cancer cell models. Arrest primarily occurred in the G2/M phase but was also noted in the G1/S phase. Likewise, SFN has been shown to modulate angiogenesis and autophagy, contributing to its broad spectrum of antiproliferative actions.

Several directions of future investigation have been identified throughout this systemic review. A substantial number of in vitro studies across cancer subtypes have identified specific anticancer properties of SFN, and it is clear that additional in vivo studies should be performed to support these mechanisms. Similarly, many in vivo studies have utilized various broccoli extracts, but because of its potent nature, it would be beneficial to understand the effects of pure SFN in animal tumor models. While BSE contains GFN that is metabolized to SFN, it is difficult to accurately control and determine the dose of SFN due to differences in gut microbes and liver conjugation enzymes. Therefore, future in vivo studies should focus on utilizing pure SFN to allow for the most accurate translation to human clinical trials.

Several anticancer mechanisms have been proposed by a variety of researchers, and some of these mechanisms contradict one another. Therefore, additional studies must be conducted to fully elucidate the molecular targets of SFN as well as identify reliable biomarkers that accurately depict the efficacy of SFN in pre-clinical and clinical studies. Additional cancer types must be explored in clinical trials as only breast, skin, pancreatic, and prostate cancer trials have been published. Furthermore, many of the reported clinical trials utilized a variety of cruciferous vegetable products; however, it would be beneficial to study the effects of pure SFN as it appears to be the phytochemical with the greatest anticancer properties. Greater sample sizes and the use of randomized controlled trials would provide more substantial support for SFN’s chemopreventive properties.

This extensive review only included in vitro and in vivo studies that reported the effects of SFN alone on different cancer cell lines. During this work, we came across many published studies (not presented here) that reported significant effects of SFN in combination with other phytochemicals and chemotherapeutic agents. Additionally, many studies have addressed the instability of SFN and have formulated a multitude of delivery systems to increase the bioavailability of SFN. Such delivery systems include microencapsulation, microspheres, nanoparticles, micelles, and liposomes. Analysis of this literature could provide additional knowledge of the anticancer and chemotherapeutic properties of SFN-containing regimens. Based on the overwhelming evidence presented in this in-depth analysis of current research, SFN is a promising antineoplastic and chemopreventive phytochemical that can be utilized as a valuable cancer-fighting agent.

## Figures and Tables

**Figure 1 cancers-13-04796-f001:**
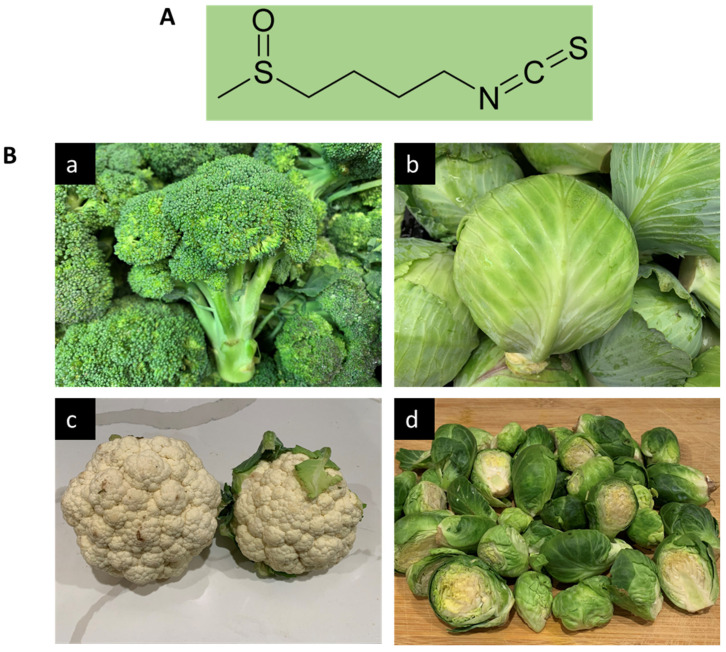
SFN and it various sources. (**A**) Molecular structure of SFN. (**B**) Dietary sources of SFN: (**a**) broccoli, (**b**) cabbage, (**c**) cauliflower, and (**d**) Brussels sprouts.

**Figure 2 cancers-13-04796-f002:**
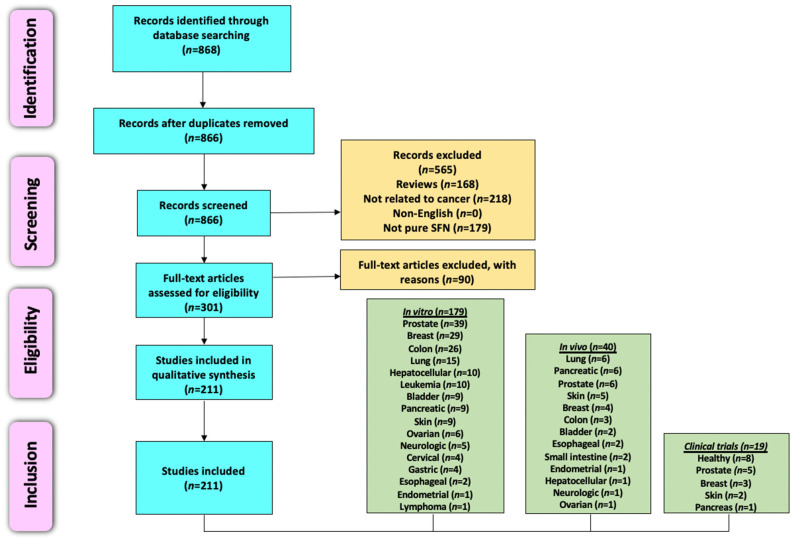
A PRISMA flow diagram depicting the literature search and study selection process relevant to anticancer potential of sulforaphane. The total number of in vitro, in vivo, and clinical studies (238) is greater than the number of studies included in this work (211) because numerous publications contained results from more than one organ-specific cancer or study type (i.e., in vitro, in vivo, or clinical).

**Figure 3 cancers-13-04796-f003:**
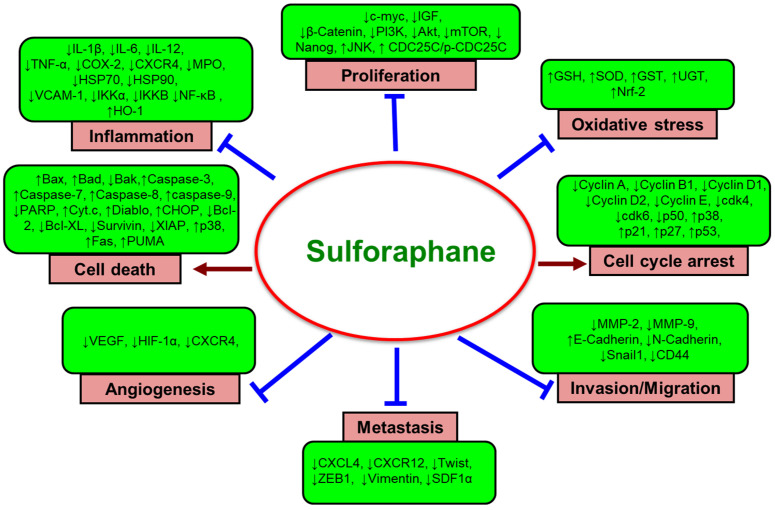
Chemopreventive and anticancer and effects of SFN extrapolated from in vitro and in vivo literature analysis. Symbols: ↑, increased or upregulated; ↓ decreased or downregulated; ⊥, blocked.

**Figure 4 cancers-13-04796-f004:**
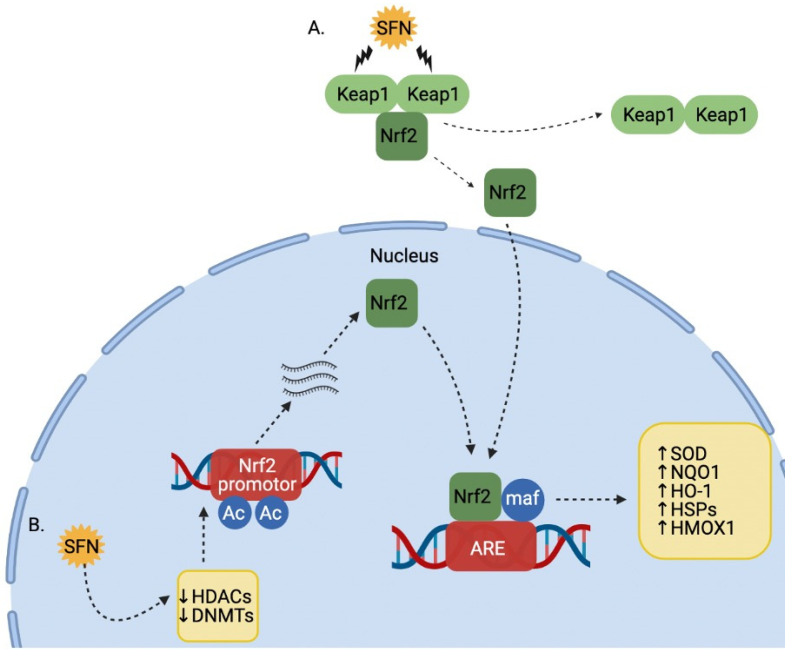
Effect of SFN on Nrf2/Keap1 pathway. (**A**) SFN modifies Keap1 cysteine residues, causing release of Nrf2, which allows it to translocate to the nucleus. (**B)** SFN also induces epigenetic modulation of HDACs and DNMTs, causing increased Nrf2 transcription and translation. Consequently, the increased nuclear Nrf2 binds to ARE and maf to increase transcription of various cytoprotective phase II enzymes. This figure was created using resources available at BioRender.com (accessed on 4 July 2021). Symbols: ↑, increased or upregulated; ↓ decreased or downregulated.

**Figure 5 cancers-13-04796-f005:**
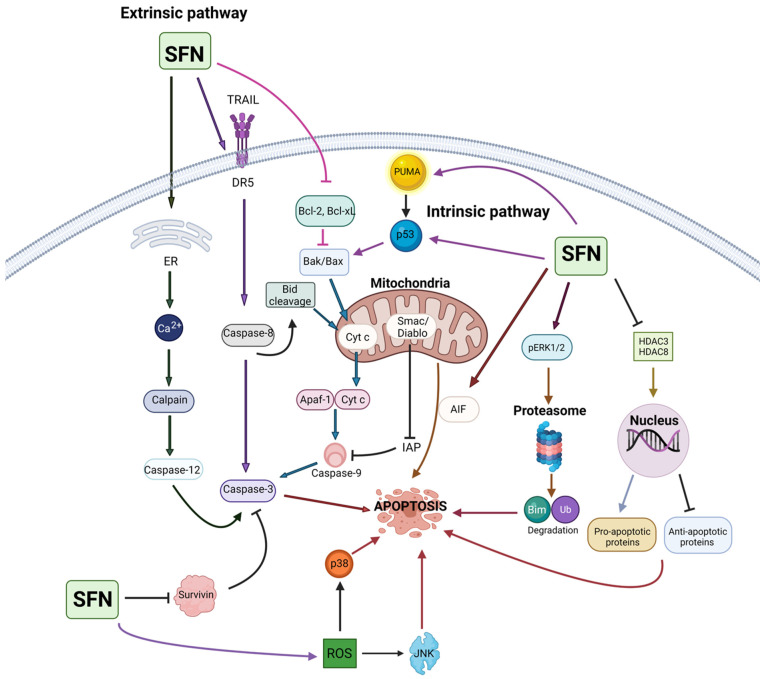
SFN modulates multiple targets in the intrinsic and extrinsic apoptotic pathways. SFN has been found to alter the expression or activate/inactivate various apoptotic mediators and regulators. Symbols: →, increased, upregulated or activated; ⊥, blocked or suppressed. This figure was created using resources available at BioRender.com (accessed on 17 September 2021).

**Table 1 cancers-13-04796-t001:** Potential antineoplastic effects and underlying mechanisms of action of SFN based on in vitro studies.

Cell Lines Used	Conc. and Duration	Anticancer Effects	Mechanisms	References
*Breast cancer*				
MCF-7	0.1–100 µM (1–48 h)	Increased cytotoxicity	Not reported	Tseng et al., 2004 [93]
MCF-7	5–30 µM (6–24 h)	Suppressed cell proliferation	↑G2/M phase arrest; ↑cyclin B1; ↑H1 phosphorylation; ↓tubulin polymerization	Jackson and Singletary, 2004 [94]
F3II	5–30 µM (12–48 h)	Inhibited cell growth	↑G2/M phase arrest; ↑cdc2 kinase activity; ↓tubulin polymerization; ↑apoptosis; ↓Bcl-2; ↓PARP; ↑caspase-3-like activity	Jackson and Singletary, 2004 [95]
MCF-7	2.5–50 µM (20–72 h)	Inhibited cell proliferation	↑G2/M phase arrest; ↑microtubule dysfunction; ↑tubulin acetylation; ↓tubulin polymerization	Azarenko et al., 2008 [96]
MDA-MB-231, MDA-MB-468	5–50 µM (3–24 h)	Inhibited cell growth and invasion	↑Apoptosis; ↑USP14; ↑UCHL5; ↑Ub-Prs	Ahmed et al., 2018 [97]
MDA-MB-231, MDA-MB-468, BT-474, MCF-7	1–25 µM (24, 72 h)	Decreased cell growth	↓HDAC5; ↓HDAC5; ↓USF1; ↓USF1; ↓luciferase; ↓LSD1; ↑H3K4me1/2; ↑ AcH3K9; ↓USP28; ↑*CTDSPL*; ↑*GLPR1*; ↑*CYLD*; ↑*TFP12*; ↑*PPP2R1B*; ↑*ISG15*; ↑*EGLN3*	Cao et al., 2018 [98]
MCF-7, MDA-MB-231	5 µM (24–72 h)	Inhibited cell growth	↑G2-M phase arrest; ↓*CCND1*; ↓*CDK4*; ↓HDAC2; ↓HDAC3; ↓HMT; ↑p53; ↑p21; ↑H3K4Me3	Royston et al., 2018 [99]
MDA-MB-231, MCF-7, T47D, MDA-MB-468	5–25 µM (24–72 h)	Inhibited cell growth	↑G2-M phase arrest; ↓cyclin B1 (MDA-MB-231 and MCF-7); ↑apoptosis; ↓global HDAC; ↓EGFR; ↓HER-2	Pledgie-Tracy et al., 2007 [100]
MCF-7, MDA-MB-231, SK-BR-3	5–20 µM (24 h)	Induced cytotoxicity	↑p21; ↑oxidative stress; ↑carbonylation of lamin A/C; ↓lamin B1; ↓nucleolar RRN3; ↑nuclear RRN3; ↓NOP2; ↓WDR12	Lewinska et al., 2017 [101]
MCF-7, MDA-MB-231, SK-BR-3	2.5–20 µM (24 h)	Inhibited cell proliferation	↑Apoptosis; ↑G2/M phase arrest (MCF-7 and MDA-MB-231); ↑G0/G1 phase arrest (SK-BR-3); ↑p53; ↑p21; ↓CCNA2; ↓CCNB1; ↓CCNB2; ↓CCND3; ↓CCNE1, CCND1; ↓CCND2; ↓CCNH; ↓p-ERK1/2 (MDA-MB-231); ↑ROS; ↑DNA DSBs; ↑DNA SSBs; ↓Akt signaling; ↓ATP; ↓AMPK activation; ↓5-mdC; ↑HDAC5; ↓HDAC6-10; ↓DNMT1; ↓DNMT3B	Lewinska et al., 2017 [102]
MCF-7, MDA-MB-231	1–100 µM (24–72 h)	Induced cell death	↑Apoptosis; ↑S phase cells; ↑G2/M phase cells; ↑p21; ↑p27; ↓cyclin A; ↓cyclin B1; ↓CDC2; ↑caspase-3; ↓Bcl-2; ↑autophagy; ↑LC3-I; ↑LC3-II	Kanematsu et al., 2010 [103]
MCF-7, SK-BR-3, MDA-MB-231, MDA-MB-468	5–30 µM (24 h)	Inhibited cell proliferation	↑Autophagosomal vacuoles; ↓mTOR; ↓S6K1; ↓p-Akt	Pawlik et al., 2013 [104]
MDA-MB-231, BT549, MDA-MB-468	10, 25 µM (16–72 h)	Inhibited cell growth	↑Autophagy; ↓P62; ↑Beclin1; ↑LC3-II; ↓HDAC6; ↑PTEN; ↓Akt	Yang et al., 2018 [105]
MDA-MB-231, MDA-MB-436, MDA-MB-468, MDA-MB-453	1–60 µM (24–72 h)	Inhibited cell proliferation	↑Apoptosis; ↑G2/M phase arrest; ↑*Egr1;* ↑*NQO1*; ↑*SL7A11*; ↑*G6PD*; ↑*GCLM*; ↑*SCD*; ↑*ID1*; ↑*IGFBP3*; ↓cyclin B1; ↓Cdc2; ↓p-Cdc2; ↓Cdc25c	Yang et al., 2016 [106]
ZR-75-1	6.25–25 µM (4–72 h)	Inhibited cell growth	↑G1/S phase arrest; ↓CDK2; ↓CDK4; ↓CDK6; ↓CDK2; ↓CDK4; ↓SERTAD1; ↓CCDN2; ↓HDAC3; ↓SERTAD; ↓CCDN2; ↓HDAC3	Cheng et al., 2019 [107]
SH, SHR	5–20 µM (72 h)	Decreased cell proliferation	↑Apoptosis; ↑S-phase arrest (SH cells); ↑S-phase and G2/M-phase arrest (SHR cells); ↓HDAC1; ↑global histone H3 acetylation; ↑DCBLD2; ↓*Septin 9*	Li et al., 2016 [108]
MCF-7, ZR-75-1	2.5–30 µM (24–72 h, 7 days)	Decreased cell proliferation	↓ERα; ↓PR; ↑PSMB5	Ramirez and Singletary, 2009 [109]
T47D, MCF-7, BT-474	2–50 µM (96 h)	Decreased cell viability	↑Apoptosis; ↑PARP cleavage	Pawlik et al., 2016 [110]
MCF-7	25 µM 24–72 h	Inhibited cell proliferation	↑Apoptosis; ↑caspase-7; ↑PARP cleavage; ↓Bcl-2; ↑Bax; ↓ERK1/2 MAPK; ↑p38 MAPK	Jo et al., 2007 [111]
MCF-7	0.01–75 µM (24 and 48 h)	Decreased cell viability	↑Apoptosis; ↓Bcl-2; ↓COX-2	Hussain et al., 2013 [112]
MCF-7, MDA-MB-231	5–20 µM (72 h)	Decreased cell viability	↑Apoptosis; ↓CYP1A1 protein (all cell lines); ↓CYP19 (MCF-7); ↑CYP19; ↑aromatase (MDA-MB-231); ↑CYP1A2 (MDA-MB-231)	Licznerska et al., 2015 [113]
MCF-7, MDA-MB-231	10–70 µM (96 h)	Inhibited cell growth	↑Apoptosis; ↑p53 (MCF-7); ↓PTEN methylation; ↑RARβ2; ↑p21	Lubecka-Pietruszewska et al., 2015 [114]
MCF-7, MDA-MB-231	5–20 µM (3–15 days)	Inhibited cell proliferation	↑Apoptosis; ↓*h*TERT; ↓telomerase activity; demethylation of CpGs of the CTCF binding site; ↑ac-H3; ↑ac-H3K9; ↓tri-me-H3K27; ↓tri-me-H3K9, ↓DNMT1; ↓DNMT3a	Meeran et al., 2010 [115]
MCF-7, MDA-MB-231	1–20 µM (24 h)	Inhibited cell proliferation	↑Apoptosis; ↓HSP70; ↓HSP90; ↓HSF1; ↑p53; ↑p21; ↑AIF; ↑Bax; ↑Bad; ↓Bcl-2; ↑caspase-3; ↑caspase-8; ↑caspase-9	Sarkar et al., 2012 [116]
MCF-7	10 µM (45 min, 6 h)	Inhibited cell growth	↑Nrf2; ↑NQO1; ↑HMOX1; ↑H3K9Ac:H3	Lo and Matthews, 2013 [117]
MCF-7	1–12 µM (4–24 h)		↑TrxR1	Wang et al., 2005 [118]
MDA-MB-231, MDA-MB-468, BT-549, BT-474, SKBR3, HS578T	20–60 µM (16, 24 h)	Decreased cell proliferation	↑Nrf2; ↓RON	Thangasamy et al., 2011 [119]
MCF-7, MDA-MB-231	1–20 µM (24 h)	Decreased cell viability and migration	↑CAV1; ↑CAV1; ↑condensed chromatin	Deb et al., 2014 [120]
MDA-MB-231-Luc-D3H1, JygMC(A)	2.5–20 µM (48 h)	Decreased cell proliferation	↓Primary tumorspheres; ↓secondary tumorspheres; ↓tertiary tumorspheres; ↓CR1; ↓CR3; ↓GRP78; ↓Alk4	Castro et al., 2019 [86]
*Gastrointestinal tract and associated cancers*				
*Esophageal cancer*				
OE33, FLO-1	1–12.5 µM (0.5–5 h)	Inhibited cell growth	↑Apoptosis; ↑G1 phase arrest; ↓HSP90; ↑p21	Qazi et al., 2010 [121]
EC9706, ECa109	10–60 µM (3–72 h)	Inhibited cell proliferation and induced autophagy	↑Apoptosis; ↑caspase-9; ↑LC3B-II; ↓p62; ↑Nrf2	Lu et al., 2020 [122]
*Gastric cancer*				
AGS	2.0–6.75 µM (3–24 h)	Decreased cell growth and migration	↑Apoptosis; ↑ROS; ↑Bax; ↓Bcl-2; ↑cyt c; ↑caspase-8; ↑PARP-1 cleavage; ↑SFE; ↑p-JNK; ↑p-P-38; ↓p-ERK1/2	Mondal et al., 2016 [123]
AGS	2.5–20 µM (24, 48 h)	Inhibited cell viability	↑Apoptosis; ↑G2/M phase arrest; ↑cyclin B1; ↑p53; ↑p21; ↑p-H3; ↑PARP cleavage; ↑p-AMPK; ↓cyt c; ↓MMP	Choi et al., 2018 [124]
MGC803, AGS	2–32 µM (24–48 h)	Inhibited cell proliferation	↑Apoptosis; ↑G2/M phase arrest; ↓SMYD2; ↓SMYD3 mRNA; ↓*CYR61*; ↓*MYL9*	Dong et al., 2018 [125]
AGS, MKN45	31–250 µM (48 h)	Inhibited cell growth	↑Apoptosis; ↑CDX1; ↑CDX2	Kiani et al., 2018 [126]
*Colon cancer*				
SW620	5–100 µM (24–72 h)	Decreased cell proliferation	↑Apoptosis; ↑caspase-3; ↑double-strand DNA breaks	Andělová et al., 2007 [127]
SW620	20 µM (12–48 h)	Inhibited cell proliferation	↑Apoptosis; ↑caspase-9; ↑caspase-3; ↑caspase-7; ↑ATM kinase; ↑Chk2 kinase; ↑JNK	Rudolf et al., 2009 [128]
SW480	1–20 µM (3–48 h)	Inhibited cell proliferation	↑Apoptosis; ↑caspase-3; ↑caspase-7; ↑caspase-9; ↑ERK; ↑p53; ↓Bcl-2; ↑Bax/Bcl-2 ratio; ↑ROS; ↑MDA	Lan et al., 2017 [129]
HT-29	5–30 µM (24–96 h)	Inhibited cell growth	↑Apoptosis; ↑G2/M phase arrest; ↑cyclin A; ↑cyclin B1; ↑Bax; ↑PARP cleavage	Gamet-Payrastre et al., 2000 [130]
40-16, 379.2	0.4–50 µM (10–72 h)	Decreased cell growth	↑Apoptosis; ↑PARP cleavage; ↓Pro-C9; ↓Pro-C7; ↓Bax; ↓Bcl-xL	Pappa et al., 2006 [131]
40-16	5, 10 µM (24–72 h)	Inhibited cell growth	↑PARP cleavage; ↑subG1 phase arrest	Pappa et al., 2007 [132]
HCT-116	0.5–100 µM (16–48 h)	Suppressed cell growth	↑Apoptosis; ↑histone H2A.X phosphorylation; ↑caspase-9; ↑caspase-3; ↑JNK; ↑Bid; ↑Bax; ↓Bcl-2	Rudolf and Cervinka, 2011 [133]
HCT-116, HT-29, DLD1, KM12, SNU-1040	2.5, 5 µM (24–72 h)	Inhibited cell proliferation	↑Apoptosis; ↑PARP cleavage; ↑G2/M phase arrest; ↑CDK1; ↑CDC25B; ↑MK2; ↑P38; ↑p-JNK; ↓microtubule polymerization	Byun et al., 2016 [134]
HT-29	6.25–100 µM (4–36 h)	Inhibited cell growth	↑G1 phase arrest; ↓*cyclin D1*; ↓*cyclin A*; ↓*c-myc*; ↑P21; ↑ERK; ↑JNK; ↑p38	Shen et al., 2006 [135]
CT116	5–15 µM (72 h)	Inhibited cell viability	↑Apoptosis; ↑G2 phase arrest; ↑p-SAPK; ↓c-Myc	Zeng et al., 2011 [136]
40-16	0.4–50 µM (3–48 h)	Inhibited cell proliferation	↑G2/M phase arrest (6, 12, and 24 h); ↑subG1 phase arrest (48 h); ↓GSH	Pappa et al., 2007 [137]
HT-29	15 µM (48 h)	Inhibited cell growth	↑Apoptosis; ↓p-cdc2; ↑p21; ↑G2/M phase arrest; ↑Rb phosphorylation; ↑Rb protein	Parnaud et al., 2004 [138]
WiDr	2.5–80 µM (16 h)	Inhibited cell proliferation	↑Apoptosis; ↑autophagy; ↑LC3-1; ↑LC3-II; ↓Bcl-2	Nishikawa et al., 2010 [139]
DLD-1, HCT116, LoVo	5–20 µM (24 h)	Inhibited cell proliferation	↓SKP2 mRNA; ↓SKP2 protein; ↑p27^KIP1^; ↑Akt; ↑ERK	Chung et al., 2015 [140]
Caco-2	1–50 µM (2, 24 h)	Inhibited cell viability	↑KLF4; ↑p21; ↓CDX-2; ↓KLF5; ↓AMACR	Traka et al., 2005 [141]
HCT-116, HT29	15 µM (6 h)	Inhibition of cell growth and migration	↑p53; ↓Wnt/β-catenin; ↑Nrf2; ↑*NMRAL2P*	Johnson et al., 2017 [142]
Caco-2	5–25 µM (6–36 h)	Induced autophagy	↑LC-II; ↑UGT1A1; ↑UGT1A8; ↑UGT1A10 mRNA; ↑Nrf2	Wang et al., 2014 [143]
Caco-2	0.5–20 µM (48 h)	Decreased cell viability	↑MRP2	Harris and Jeffery, 2008 [144]
HTC-116, HT-29, SW48, SW480	15 µM (24 h)	Decreased cell viability	↑G2/M phase arrest; ↓HDAC3; ↓HDAC6; ↑p-H2AX; ↑p-ATR; ↑CtIP acetylation	Rajendran et al., 2013 [145]
RKO, HCT-116	2.5–20 µM (72 h)	Inhibited cell growth	↑Apoptosis; ↓miR-21; ↓HDAC mRNA; ↓hTERT mRNA	Martin et al., 2018 [146]
HCT-116, SW480	0.9–60 µM (24 h)	Decreased cell proliferation	↑pH2AX; ↑pRPA32; ↑p300; ↑histone H4 acetylation	Okonkwo et al., 2018 [147]
HT-29	0.25–10 µM (24 h)	Inhibited cell growth	↓TNF-α; ↓IL-1β; ↓IL-6; ↓IFN-γ; ↓IL-1β	Bessler and Djaldetti, 2018 [148]
HT-29	10–50 µM (24, 48 h)	Suppressed cell growth and migration	↑Apoptosis; ↑subG1 phase arrest; ↑caspase-3; ↓COX-2; ↓HIF-1; ↓VEGF; ↓CXCR4; ↓PGE2	Tafakh et al., 2018 [149]
Caco-2	5–100 µM (24, 72 h)	Inhibited cell proliferation	↑p-ERK1/2; ↑p-Akt; ↑NQO1; ↑UGT1A1; ↓MRP2	Jakubikova et al., 2005 [150]
HCT-116, LoVo, Caco-2, HT-29	1–15 µM (72 h)	Decreased cell growth	↑Apoptosis; ↑ROS; ↓procaspase-8	Kim et al., 2010 [151]
HCT-116	12.5–50 µM (1–24 h)	Inhibited cell migration	↓HIF-1α; ↓VEGF; ↓HO-1; ↓GLUT1	Kim et al., 2015 [152]
*Hepatocellular cancer*				
HepG2, Hepa1c1c7	5–100 µM (24 h)	Exhibited cell cytotoxicity	↑ERK2; ↑MAPK pathway	Yu et al., 1999 [153]
HepG2	100 µM (24 h)	Reduced cell viability and promoted cell death	↑Apoptosis; ↑MT-I RNA; ↑MT-II RNA, ↑MT protein expression; ↑Nrf2; ↑p38; ↑JNK/MAPK pathways; ↑caspase-3; ↑PARP cleavage; ↑Bax; ↓Bcl-2; ↓Bcl-xL	Yeh and Yen, 2005 [154]
HepG2	5–30 µM (48 h)	Inhibited cell viability and promotes cell death	↑Apoptosis; ↑caspase-3; ↑Bax; ↓Bcl-2; ↓Bcl-xL; ↓PARP; ↓β-catenin	Park et al., 2007 [155]
HepG2	20 µM (24 h)	Reduced cellular proliferation	↑HO-1; ↑ARE; ↑Nrf2; ↑ERK1/2; ↓Keap1; ↓p38 MAPK; ↓p-MKK3/6	Keum et al., 2006 [156]
Huh-7, SNU-449, NCTC	20–60 µM (24 h)	Reduced cell viability and promoted cell cycle arrest	↑Apoptosis; ↑G2/M phase arrest; ↑caspase-3; ↑caspase-8; ↑caspase-9; ↓PFKFB4; ↓HIF-1α; ↓VEGF	Jeon et al., 2011 [157]
HepG2	1.25–20 µM (24 h)	Reduced cellular proliferation, adhesion, migration, and invasion	↓STAT3; ↓HIF-1α; ↓VEGF	Liu et al., 2017 [158]
Hep3B	5–20 µM (24, 48 h)	Decreased cell viability and promoted cell death	↑Apoptosis; ↓telomerase; ↓hTERT; ↓Akt; ↑ROS	Moon et al., 2010 [159]
HepG2	10–80 µM (48 h)	Inhibited cell proliferation, migration, and invasion	↑Apoptosis; ↓TGF-β–induced EMT; ↓Vimentin; ↑E-Cadherin; ↑GO/G1 arrest; ↑ROS	Wu et al., 2016 [160]
HepG2	80 µM (24–72 h)	Inhibited cell proliferation and promoted cell death	↑Apoptosis; ↑Bip/GRP78; ↑XBP-1; ↑caspase-12; ↑Bid; ↑CHOP/GADD153	Zou et al., 2017 [161]
Hepa 1c1c7, HepG2	1–40 µM (24 h)	Reduced cell viability	↑CYP1A1; ↑AhR transformation; ↑AhR binding to XRE	Anwar-Mohamed and El-Kadi, 2009 [162]
*Pancreatic cancer*				
MIA PaCa-2, PANC-1	5–40 µM (24–72 h)	Promoted cell cycle arrest and death	↑Apoptosis; ↑caspase-3, ↑caspase-8, ↑G2-M arrest, ↑ROS	Pham et al., 2004 [163]
AsPC-1, BxPc-3, MIA PaCa-2, PANC-1	0.1–100 µM (12–48 h)	Inhibited cell proliferation and promoted cell death	↑Apoptosis; ↓Akt; ↓Cdk4; ↓p53; ↑proteasomal degradation of HSP90 client proteins; ↑caspase-3; ↓HSP90-p50Cdc37 complex	Li et al., 2012 [164]
AsPC-1, BxPc-3, Capan-1, MIA PaCa2	10 µM (24, 48 h)	Reduced cell viability	↑Apoptosis; ↓NF-κB binding	Kallifatidis et al., 2009 [165]
ASPC, PANC-1, and human pancreatic CSCs	5–20 µM (1–7 days)	Reduced cellular proliferation and promoted cell death	↑Apoptosis; ↓Bcl-2, ↑caspase-3; ↓Nanog; ↓Oct-4; ↓PDGFα; ↓Smo; ↓Gli1; ↓Gli2	Rodova et al., 2012 [166]
AsPC-1, BxPc-3, PANC-1, MIA PaCa-2	10 nM (24 h)	Promoted cell death	↑Apoptosis; ↑miR-365a-3p	Yin et al., 2019 [167]
AsPC-1, BxPc-3, PANC-1	10 µM (24 h)	Promoted cell death	↑miR-135b-5p; ↑RASAL2	Yin et al., 2019 [168]
PANC-1, MIA PaCa-2	1–100 µM (24–72 h)	Inhibited cellular proliferation, invasion, and migration	↑Apoptosis; ↑ROS; ↑AMPK; ↑E-Cadherin; ↓N-Cadherin; ↓Vimentin; ↑Nrf2; ↑HO-1	Chen et al., 2018 [169]
BxPc-3, AsPC-1	10 µM (24 h)	Reduced cell viability	↑E-Cadherin; ↑GJIC; ↑Cx43; ↓c-Met; ↓CD133	Forster et al., 2014 [170]
PANC-1	10 µM (24 h)	Exhibited cytotoxicity	↑Cx43; ↑GJA1 mRNA; ↑GJIC; ↓miR30a-3p	Georgikou et al., 2020 [171]
*Gynecological cancers*				
*Cervical cancer*				
HeLa	5–30 µM (48 h)	Decreased cell viability	↑Apoptosis; ↑sub-G1 phase arrest; ↑Bax; ↓Bcl-2; ↓Bcl-xL; ↓pro-caspase-3; ↓PARP; ↓β-catenin	Park et al., 2007 [156]
HeLa	0.01–100 µM (24 h)	Inhibited cell growth	↑Apoptosis; ↑caspase-3; ↓Bcl-2; ↓COX-2; ↓IL-1β	Sharma et al., 2011 [172]
HeLa	2.5 µM (24–72 h)	Inhibited cell growth	↓DNMT; ↓DNMT3B; ↓HDAC1; ↑RARβ; ↑CDH1; ↑DAPK1; ↑GSTP1	Khan et al., 2015 [173]
HeLa	6.25–25 µM (24, 72 h)	Decreased cell proliferation	↑G2/M phase arrest; ↑MPM-2; ↓cyclin B1; ↓cyclin B1/CDC2 complex; ↑CDC25C/p-CDC25C ratio; ↑GADD45β	Cheng et al., 2016 [174]
*Endometrial cancer*				
MFE280, KLE, Ishikawa, Hec1B, Hec1A, MFE296, AN3CA	1–32 µM (24–72 h)	Inhibited cell viability	↑Apoptosis; ↑G2/M phase arrest; ↓ATP; ↑p21; ↑p27; ↑Cdc2 phosphorylation, ↑caspase-3; ↑Bax; ↓Bcl-2; ↓Cox IV; ↓MEK; ↓ERK	Rai et al., 2020 [175]
*Ovarian cancer*				
SKOV3	10–100 µM (12, 24 h)	Inhibited cell growth	↑Apoptosis; ↓Akt; ↓p-Akt; ↓PI3K; ↓cyclin D1; ↓cdk4; ↓cdk6	Chaudhuri et al., 2007 [176]
OVCAR3, SKOV3	2–50 µM (24–72 h)	Reduced cell proliferation	↑Apoptosis; ↑G1 phase arrest	Chuang et al., 2007 [177]
MDAH 2274, SKOV3	5–20 µM (12–72 h)	Induced growth arrest and inhibited cell migration	↑Apoptosis; ↑G1 phase arrest; ↓RB; ↓p130; ↑p107; ↓E2F-1; ↓E2F-2; ↓E2F-3; ↓G1 phase; ↓cyclins; ↓CDKs; ↑non-phosphorylated RB; ↓E2F-1	Bryant et al., 2010 [178]
OVCAR3, OVCAR4, OVCAR5, SKOV3	100 µM (72 h)	Inhibited cell proliferation	↑p38; ↑ERK; ↑JNK (OVCAR3 and SKOV3); ↑thioredoxin reductase (OVCAR3)	Kim et al., 2017 [179]
A2780, SKOV3	2–200 µM (6, 24 h)	Decreased cell growth	↑Apoptosis; ↑HSP27; ↑JNK; ↑MEK1; ↑p38; ↑p90^rsk^ phosphorylation; ↑IP_3_R2; ↑NFR2; ↑CHOP; ↑ATF4; ↑GCLC; ↑HMOX1; ↑NQO-1	Hudecova et al., 2016 [180]
PA-1	6.25, 12.5 µM (24, 72 h)	Inhibited cell proliferation	↑G2/M phase arrest; ↓CDC2; ↓cyclin B1/CDC2 complex	Chang et al., 2013 [181]
*Hematological cancers*				
*Leukemia*				
Jurkat T	3–30 µM (24–72 h)	Decreased cell proliferation	↑Apoptosis; ↑G2/M phase arrest	Fimognari et al., 2002 [182]
Jurkat T	3–30 µM (24, 48 h)	Reduced cell viability	↑Apoptosis; ↑G2/M phase arrest; ↑p53; ↑Bax; ↓cyclin D3; ↓CDK4; ↓CDK6	Fimognari et al., 2003 [183]
HL60	10–110 µM (24–72 h)	Inhibited cell viability	↑Apoptosis	Fimognari et al., 2008 [184]
U937	1–5 µM (48 h)	Reduced cell growth	↑Apoptosis; ↑sub-G1 phase arrest; ↑Bax; ↓Bcl-2; ↑caspase-3; ↑PARP cleavage; ↑ROS; ↑MMP	Choi et al., 2008 [185]
U937, HL60, NB-4, KG-1	15–60 µM (24, 48 h)	Decreased cell proliferation	↑Apoptosis; ↓miR-155	Koolivand et al., 2018 [186]
HL60	6–10 µM (10–48 h)	Decreased cell viability	↑G2/M phase arrest; ↑ROS ↑intracellular Ca^2+^; ↑caspase-3; ↑caspase-8; ↑caspase-9; ↑Bax; ↑Bid; ↑Fas; ↑Fas-L; ↑Endo G; ↑AIF; ↑cyt c; ↓Bcl-xL; ↑FADD	Shang et al., 2016 [187]
HL60	1–25 µM (24, 48 h)	Inhibited cell viability	↑Apoptosis; ↑NQO1; ↓NQO1; ↓Keap1; ↑Nrf2; ↓PARP; ↓pro-caspase-2; ↓pro-caspase-3; ↓p50; ↑Bax; ↓Bcl-2; ↓NF-κB	Wu et al., 2016 [188]
Nalm-6, REH, RS4	2–40 µM (24, 48 h)	Reduced cell growth	↑Caspase-3; ↑caspase-8; ↑caspase-9; ↑PARP cleavage; ↑G2/M phase; ↑S phase arrest; ↑p21; ↑cyclin B1; ↓Akt; ↓p-mTOR	Suppipat et al., 2012 [189]
B1647	10, 30 µM (24 h)	Inhibited cell proliferation	↓AQP8; ↓ROS; ↑Nox4	Prata et al., 2018 [190]
L-1210	1–5 µM (24, 48 h)	Induced cell growth arrest	↑Apoptosis; ↑DNA strand breaks; ↑PS externalization	Misiewicz et al., 2003 [191]
*Lymphoma*				
B-lymphoma cells	1–10 µM (24 h)	Inhibited cell growth	↑Caspase-3; ↑caspase-7; ↑caspase- 9; ↑PARP cleavage; ↓p38 MAPK; ↓Akt	Ishiura et al., 2019 [192]
*Lung cancer*				
A549	1–100 µM (24 h)	Induced mitotic arrest and promoted cell death	↑Apoptosis; ↑G1/S arrest; ↑G2/M arrest; ↓tubulin polymerization; ↑ROS; ↓GSH	Mi and Chung, 2008 [193]
LTEP-A2	6.25–50 µM (3–72 h)	Inhibited cellular proliferation	↑Apoptosis; ↑G2/M arrest	Liang et al., 2008 [194]
A549	30–90 µM (24 h)	Decreased cell proliferation	↑G2/M phase; ↓G0/S phase; ↑p21; ↓cyclin D1	Zuryn et al., 2016 [195]
H1299	5–15 µM (24, 48 h)	Promoted cell cycle arrest and decreased cell viability	↑Apoptosis; ↑necrosis; ↑G2/M phase; ↓G0/S phase; ↓cyclin B1; ↑cyclin D1; ↑cyclin K	Zuryn et al., 2019 [196]
A549, H1299	5–15 µM (48 h)	Promoted cell cycle arrest and cell death	↑Apoptosis; ↑H3 acetylation; ↑H4 acetylation; ↑p53; ↑p21; ↑Bax; ↑G0/G1 arrest; ↑G2/M arrest; ↓HDAC	Jiang et al., 2016 [197]
A549	2.5, 5 µM (5 days)	Decreased cell viability	↑H3K4me1; ↓miR-9-3; ↓DNMT3a; ↓HDAC1; ↓HDAC3; ↓HDAC6; ↓CDH1; ↓CpG methylation	Gao et al., 2018 [198]
A549, H1299	1–15 µM (7 days)	Inhibited cell proliferation and the formation of tumorspheres	↑Apoptosis; ↓miR-19a; ↓miR-19b; ↓Wnt/β-catenin pathway; ↑Bax; ↑caspase-3; ↑caspase-8; ↑caspase-9; ↓CD133; ↓CD44; ↓ALDH1A1; ↓nanog; ↓oct4; ↓PCNA; ↓cyclin D1	Zhu et al., 2017 [199]
H1299, 95C, 95D	0.5–100 µM (24, 48 h)	Inhibited cell proliferation, migration, and invasion	↓miRNA-616-5p; ↓β-catenin; ↓N-cadherin; ↓vimentin	Wang et al., 2017 [200]
A549, CL1-5	10–40 µM (72 h)	Reduced cell viability and aggregation	↑Apoptosis; ↑chromatin condensation; ↑anoikis, ↑annexin V binding; ↑PS externalization; ↑p53; ↑p21; ↑Bad; ↑Bax; ↑cleaved PARP, ↓procaspase-3; ↓procaspase-7; ↓procaspase-9; ↓p-FAK; ↓p-Akt; ↓β-catenin	Tsai et al., 2019 [201]
XWLC-05	0.5–5 µg/L (24, 48, 72 h)	Promoted cell cycle arrest and death	↑Apoptosis; ↑G2/M phase; ↓G0/S phase; ↑p73; ↑PUMA; ↑Bax; ↑caspase-9; ↓Bcl-2; ↓p53	Zhou et al., 2017 [202]
Cadmium-transformed BEAS-2BR	2.5, 5, 10 µM (24 h)	Exhibited cytotoxicity	↑Apoptosis; ↓apoptosis resistance; ↑autophagy; ↑caspase-3; ↑C-PARP; ↓constitutive Nrf2; ↓Bcl-2	Wang et al., 2018 [203]
PC9/gef, H1975, A549, CL1-5, H3255	5–20 µM (48 h)	Reduced cell proliferation	↓pEGFR; ↓p-Akt; ↓p-STAT3; ↑proteasome activity	Chen et al., 2015 [204]
A549, H1299	0.5–5 µM (72 h)	Reduces cellular proliferation, migration, and invasion	↑ERK5; ↑p-ERK5 ↑E-Cadherin; ↑ZO-1; ↓pc-jun; ↓pc-Fos; ↓N-Cadherin; ↓Snail1; ↓MMP-2	Chen et al., 2019 [205]
SK-1, A549	5–30 µM (24 h)	Decreased cell viability and promoted cell death	↑Apoptosis; ↑ERK1/2; ↑Bax; ↑caspase-3; ↑26S proteasome activity; ↓Bim	Geng et al., 2017 [206]
HBE exposed to 2% TS and A549	1–40 µM (1–7 days)	Inhibited TS-induced, CSC-like properties	↓CD133; ↓ALDH1A1; ↓Oct4; ↓Nanog; ↓IL-6; ↓NICD; ↓Hes1; ↓ΔNp63α	Xie et al., 2019 [207]
*Neurological cancer*				
T98G and U87MG	20, 40 µM (24, 48 h)	Decreased cell viability and promoted cell death	↑Apoptosis; ↑intracellular Ca^+2^; ↑Bax:Bcl2; ↑caspase-3; ↑caspase-9; ↑caspase-12; ↑cyt. c; ↑calpain; ↑α-spectrin degradation; ↑ICAD cleavage; ↑AIF; ↑Smac; ↑Diablo; ↓IAPs; ↓NF-κB	Karmakar et al., 2006 [208]
U87, U373, U118, SF767	5–50 µM (24, 48 h)	Inhibited cell survival and promoted cell death	↑Apoptosis; ↑ROS; ↑DNA double-strand breaks; ↑γ-H2AX ↑caspase-3; ↑caspase-7; ↑caspase-9	Bijangi-Visheshsaraei et al., 2017 [209]
U87 and U251	1–50 µM (24, 48 h)	Reduced cell viability and promoted cell death	↑Apoptosis; ↑caspase-3; ↑Bax; ↑ROS; ↓Bcl-2; ↓p-STAT3	Miao et al., 2017 [210]
U251MG	10–40 µM (24 h)	Reduced cell viability and invasion	↑Apoptosis; ↑Bad; ↑Bax; ↑cyt. c; ↑Annexin V-binding capacity; ↓Bcl-2; ↓survivin; ↓invasion; ↓MMP-2; ↓MMP-9; ↓Galectin-3	Zhang et al., 2016 [211]
U87MG, U373MG	10–90 µM (24 h)	Decreased cell proliferation, migration, and invasion	↑ERK1/2; ↑CD44v6; ↓MMP-2	Li et al., 2014 [212]
*Skin cancer*				
ME-18	1–5 µM (24, 48 h)	Induced cell growth arrest	↑Apoptosis; ↑DNA strand breaks; ↑PS externalization	Misiewicz et al., 2003 [191]
A375, 501MEL	1–5 µg/mL (2–48 h)	Suppressed cell growth, invasion, and metastasis	↑Apoptosis; ↑MDM2; ↑BAX; ↑PUMA; ↑GADD45A; ↓CDKN1A; ↑FAS; ↑caspase-3; ↑caspase-8; ↑caspase-9; ↓Bcl2; ↑BBC3; ↓ADORA1; ↑HMOX1; ↑TXNRD1; ↑GGLC; ↑GCLM;↑AKR1B10; ↑G6PD; ↑HTRA3; ↓FST; ↓ITGB4; ↓PLAT; ↓ITGB2; ↓G2/M phase; ↑CDKN1A; ↑EGR1; ↑GADD45B; ↑ATF3	Arcidiacono et al., 2018 [83]
A375	2 µg/mL (24–72 h)	Shifted growth factor receptor ratio from prosurvival to proapoptotic	↑Apoptosis	Arcidiacono et al., 2018 [213]
A375	0.1–100 µM (24, 48 h)	Decreased cell survival	↑Apoptosis; ↑caspase-3; ↑caspase-4; ↑caspase-6; ↑caspase-7; ↑caspase-8; ↑caspase-9	Mantso et al., 2016 [214]
A375 and WM793	1–20 µM (24, 48 h)	Reduced spheroid formation, migration, and invasion	↑Apoptosis; ↓Ezh2; ↓H3K27me3; ↓Bmi-1; ↓Suz12	Fisher et al., 2016 [215]
B16F-10	1–5 µg/mL	Reduced cell viability and proliferation	↑Apoptosis; ↑caspase-3; ↑caspase-9; ↑Bax; ↑p53; ↓caspase-8; ↓Bcl-2; ↓Bid; ↓NF-κB; ↓IL-1β; ↓IL-6; ↓TNF-α; ↓IL-12p40; ↓GM-CSF; ↓p65; ↓p50; ↓c-Fos; ↓ATF-2; ↓CREB; ↓c-Rel	Hamsa et al., 2011 [216]
B16	20–50 µM (24–72 h)	Reduced cell viability	↓HDAC	Enriquez et al., 2013 [217]
B16 and S91	20–50 µM (24–72 h)	Inhibited cell growth and proliferation	↓HDAC	Do et al., 2010 [218]
Bowes and SK-MEL-28	5–100 µM (2–48 h)	Decreased cellular proliferation	↑Apoptosis; ↑p-p38 kinase; ↑p53; ↑PUMA; ↑Bax; ↑ROS	Rudolf et al., 2014 [219]
*Urogenital cancers*				
*Bladder cancer*				
T24	5–20 µM (24, 48 h)	Inhibited cell proliferation	↑Apoptosis; ↓S and G2/M phase cells; ↑p27	Shan et al., 2006 [220]
T24	50–20 µM (4–24 h)	Decreased cell growth	↓COX-2; ↑nuclear NF-κB translocation; ↑p38	Shan et al., 2009 [221]
T24	5–20 µM (10, 24 h)	Inhibited cell growth	↑TR-1 mRNA; ↑GSTA1 mRNA; ↓COX-2	Shan et al., 2010 [222]
T24	5–20 µM (24 h)	Decreased cell invasion and migration	↑E-cadherin; ↓Snail; ↓ZEB1; ↑miR200c	Shan et al., 2013 [223]
T24	10, 20 µM (24 h)	Inhibited cell growth	↑Apoptosis; ↑caspase-3; ↑caspase-9; ↑PARP cleavage; ↓XIAP; ↓cIAP-1; ↓cIAP-2; ↑Bax; ↑cyt. c; ↑ER stress; ↑*GRP78;* ↑*CHOP*; ↑ROS; ↑Nrf2; ↓Keap1; ↑HO-1	Jo et al., 2014 [224]
RT4, J82, UMUC3	5–100 µM (48 h)	Inhibited cell proliferation	↑Apoptosis; ↓NHU; ↑G2/M phase arrest; ↑caspase-3; ↑caspase-7 activity; ↑PARP cleavage; ↓survivin; ↓EGFR; ↓HER2/neu	Abbaoui et al., 2012 [82]
RT4, J82, UMUC3	4–20 µM (3–48 h)	Decreased cell growth	↓HDAC; ↑p21 (RT4 cells); ↓thymidylate synthase; ↓histone H1 phosphorylation; ↑PP1β; ↑PP2A	Abbaoui et al., 2017 [225]
BIU87	10–80 µM (24 h)	Decreased cell proliferation	↑Apoptosis; ↑G2/M phase arrest; ↑IGFBP-3 mRNA; ↓NF-κB	Dang et al., 2014 [226]
5637	20 µM (4–48 h)	Suppressed cell growth	↑Apoptosis; ↑G2/M phase arrest; ↑histone H3 phosphorylation; ↑cyclin B1; ↑Cdk1; ↑caspase-3; ↑PARP cleavage; ↑MMP loss; ↑ROS	Park et al., 2014 [227]
*Prostate cancer*				
LNCaP, MDA PCa 2a, MDA PCa 2b, PC-3, TSU-Pr1	0.1–0.15 µM (1–72 h)	Reduced cellular proliferation	↑NQO1; ↑QR; ↑γ-GCS-L; ↑GSH; ↑microsomal GST; ↑α-class GSTs	Brooks et al., 2001 [228]
DU145, LNCaP, PC-3, and CWR22Rv1	20, 40 µM (12, 24 h)	Reduced cell viability	↑Apoptosis; ↓p-STAT3; ↓IL-6-induced STAT3 phosphorylation; ↓JAK2; ↓pSTAT3 nuclear translocation; ↓STAT3 dimerization; ↓Bcl-2; ↓cyclin D1; ↓survivin; ↓Mcl-1	Hahm et al., 2010 [229]
LNCaP, PC-3	10–40 µM (2–24 h)	Reduced cell growth and proliferation	↑Apoptosis; ↑p53; ↑Bax; ↑E2F1; ↑Apaf-1; ↓Bak; ↓Bcl-xL; ↓NF-κB; ↓cIAP1; ↓cIAP2; ↓XIAP	Choi et al., 2007 [230]
LNCaP	1, 10 µM (24–72 h)	Reduced cell viability and growth	↑Apoptosis; ↓Bcl-xL; ↓glycolysis; ↓HIF-1α; ↓nuclear AR; ↓PSA	Carrasco-Pozo et al., 2019 [231]
LNCap, C4-2	1–40 µM (24, 48 h)	Inhibited androgen-stimulated cell growth and proliferation	↑Transcriptional repression of AR; ↓total AR; ↓Ser210/213 phosphorylated AR; ↓intracellular PSA; ↓secreted PSA	Kim and Singh, 2009 [232]
PC-3, LNCaP	40 µM (16 h)	Reduced cell viability and promoted cell death	↑Apoptosis; ↑autophagy; ↑LC3; ↑cyt. c	Herman-Antosiewicz et al., 2006 [233]
LNCaP, PC-3	20 µM (24 h)	Inhibited cell growth and proliferation	↑Apoptosis; ↑autophagy; ↑LC3 cleavage; ↑ROS; ↑G2/M phase arrest; ↑cyt. c; ↑Bax; ↓Bcl-2; ↓respiratory chain activity	Xiao et al., 2009 [234]
LNCaP, PC-3	150, 300 µM (4 h)	Decreased cell proliferation	↑Apoptosis; ↑autophagy; ↑LC3-II; ↓p62	Watson et al., 2015 [235]
DU145	5–20 µM (24, 48 h)	Inhibited cell viability	↑Apoptosis; ↑G2/M phase arrest; ↑PARP cleavage; ↑ROS; ↑JNK	Cho et al., 2005 [236]
PC-3	10–40 µM (24 h)	Reduced cell viabilityand proliferation	↑Apoptosis; ↑DNA double-strand breaks; ↑S-phase arrest	Hac et al., 2020 [237]
LNCaP	20–100 µM (24 h)	Decreased cell viability and growth	↑Apoptosis; ↑PARP cleavage; ↑caspase-3; ↓PGM3	Lee et al., 2010 [238]
PC-3	20–100 µM (24–72 h)	Reduced cell survival and proliferation	↑Apoptosis; ↑G0/G1 arrest; ↑caspase-3; ↑caspase-8; ↑caspase-9; ↑Bax; ↑PARP cleavage; ↓Bcl-2	Singh et al., 2004 [239]
PC-3, DU145	10–40 µM (1–24 h)	Reduced cell viability	↑Apoptosis; ↑caspase-3; ↑caspase-9; ↑Bid cleavage; ↑PARP cleavage; ↑Fas; ↑cyt. c; ↑disruption of mitochondrial membrane potential; ↑ROS; ↓GSH	Singh et al., 2005 [240]
PC-3	5–20 µM (24–96 h)	Reduced cell viability	↑Apoptosis; ↑NRF1; ↑mitochondrial fission; ↑Bax; ↑PGC1α; ↓HIF-1α	Negrette-Guzmán et al., 2017 [241]
22Rv1	5–50 µM (3–24 h)	Inhibited cell growth	↑Apoptosis; ↓USP14 and UCHL5 active sites; ↑USP14; ↑UCHL5 protein; ↑Ub-Prs	Ahmed et al., 2018 [97]
LNCaP, PC-3	15 µM (6, 24 h)	Inhibited cell proliferation	↑HO-1; ↑NQO1; ↓BMX; ↓CDK2; ↓PLK1; ↓Sp1	Beaver et al., 2014 [242]
LNCaP	10, 25 µM (2–72h)	Reduced cell growth	↑Apoptosis; ↑NQO1; ↑LTB4DH; ↑ME1; ↑TXNRD1; ↑GSTM1; ↑MGST1; ↑SOD1; ↑PRDX1; ↑GCLM; ↓Jun; ↑G2/M arrest	Bhamre et al., 2009 [243]
LNCaP, PC-3	15 µM (24, 48 h)	Inhibited cellular growth	↑Ac-H3 at P21 promoter; ↑p21; ↑G2/M phase arrest; ↑HO-1; ↑NQO1; ↓HDAC3; ↓HDAC4; ↓HDAC6	Clarke et al., 2011 [71]
LNCaP, VCaP	10–20 µM (12, 24 h)	Decreased cell viability	↑HSP90 acetylation; ↓AR; ↓HDAC6; ↓ERG	Gibbs et al., 2009 [244]
LNCaP, PC-3	15 µM (48 h)	Promoted cell cycle arrest and death	↓HDAC activity; ↑Ac-H3; ↑Ac-H4; ↑caspase-3; ↑G2/M arrest	Myzak et al., 2006 [245]
TRAMP C1		Exhibited cytotoxicity	↑Nrf2; ↑NQO-1; ↑Ac-H3; ↓DNMT1; ↓DNMT3a; ↓HDAC1; ↓HDAC4; ↓HDAC5; ↓HDAC7	Zhang et al., 2013 [246]
LNCaP, PC-3	15 µM (3–24 h)	Decreased cellular proliferation	Altered ~100 lncRNA’s expression	Beaver et al., 2017 [247]
PC-3, LNCaP	10, 20 µM (8–24 h)	Reduced cell proliferation and migration	↓Notch1; ↓Notch2; ↓Notch4; ↑DNA fragmentation	Hahm et al., 2012 [248]
PC-3, LNCaP	15, 30 µM (24, 48 h)	Decreased cellular proliferation	↓DNMT1; ↓DNMT3b; ↓cyclin-D2-promoter methylation; ↑cyclin D2	Hsu et al., 2011 [249]
LNCaP, PC-3	15 µM (48 h)	Exhibited cytotoxicity	↓DNMT1; ↓DNMT3b; ↑CCR4; ↑TGFBR1	Wong et al., 2014 [250]
LNCaP, PC-3	2.5–20 µM (24 h)	Decreased cell viability and proliferation	↑Apoptosis; ↓pCSC; ↓CD24; ↓ITGA6; ↓ZEB2; ↓c-Myc	Vyas et al., 2016 [251]
LNCaP, PC-3	15 µM (6–24 h)	Decreased cell viability	↑SUV39H1 post-translational modification; ↓H3K9me3; ↓chromatin-associated SUV39H1	Watson et al., 2014 [252]
LNCaP, 22Rv1, PC-3	5, 10 µM (24 h)	Decreased cell viability	↓Glycolysis; ↓HKII; ↓LDHA; ↓PMK2	Singh et al., 2019 [253]
DU145	5–40 µM (24 h)	Decreased cell viability, migration, and invasion	↓Pseudopodia; ↓MMP-2; ↑p-ERK1/2; ↑E-Cadherin; ↓CD44v6	Peng et al., 2015 [254]
PC-3, DU145	10, 20 µM (24 h)	Decreased cell proliferation and migration	↑Apoptosis; ↑Vimentin; ↑PAI-1; ↓E-cadherin	Vyas and Singh, 2014 [255]
PC-3	40 µM (3–24 h)	Inhibited cell viability	↑Autophagy; ↓S6K1 phosphorylation; ↑LC3	Hac et al., 2015 [256]
PC-3	5–50 µM (24 h)	Decreased cell viability	↑H2S; ↑p38; ↑JNK	Pei et al., 2011 [257]
PC-3	10–40 µM (2–24 h)	Decreased cell survival	↓Protein synthesis; ↓[3H]-leucine incorporation; ↓mTOR signaling; ↑S6K1 dephosphorylation; ↓survivin	Wiczk et al., 2012 [258]
PC-3	1–40 µM (24 h)	Decreased cell viability	↓NF-κB; ↓p65 nuclear translocation; ↓VEGF; ↓cyclin-D1; ↓Bcl-xL; ↓IKKα phosphorylation; ↓IKKβ phosphorylation	Xu et al., 2005 [259]
PC-3	5–40 µM (6–24 h)	Reduced cell viability	↑AP-1; ↑p-ERK1/2; ↑p-JNK1/2; ↑p-Elk-1; ↑p-c-Jun	Xu et al., 2006 [260]
DU145	5–40 µM (24 h)	Inhibited angiogenesis	↓HIF-1α; ↑JNK signaling; ↑ERK signaling; ↓VEGF	Yao et al., 2008 [261]
LNCaP, 22Rv1	5, 10 µM (8–24 h)	Inhibited cell proliferation	↓ACC1; ↓FASN; ↓CPT1A; ↓ACADVL; ↓ACADM; ↓HADHA; ↓SREBP1	Singh et al., 2018 [262]
LNCaP	10–60 µM (24 h)	Promoted cell cycle arrest and death	↑G2/M arrest; ↑S phase arrest; ↑mitotic arrest; ↓cyclin D1; ↓cyclin E1; ↓Cdk4; ↓Cdk6; ↓Cdk1; ↓Cdc25C; ↑cyclin B1; ↑p53; ↑p21	Herman-Antosiewicz et al., 2007 [263]
LNCaP, DU-145	15 µM (24 h)	Reduced cellular proliferation	↓hTERT; ↓G0/G1 transition; ↓S phase; ↓NF-κB; ↓HDAC inhibitor activity; ↓H3K4me2 signal; ↓MeCP2; ↑H3K18Ac signal ↑DNMT1; ↑DNMT3a; ↑Pan-acetylated H3; ↑Pan-acetylated H4	Abbas et al., 2016 [264]

Symbols: ↑, increased or upregulated; ↓decreased or downregulated.

**Table 2 cancers-13-04796-t002:** Potential antineoplastic effects and underlying mechanisms of action of SFN based on in vivo studies.

Animal Tumor Models	Anticancer Effects	Mechanisms	Dose (Route)	Duration	References
*Breast cancer*					
BALB/c mice injected with F3II cells	Suppressed tumor development	↓PCNA; ↑PARP fragment	15 nmol, daily (i.v.)	13 days	Jackson and Singletary, 2004 [95]
Nude female BALB/c mice xenografted with MDA-MB-231-Luc-D3H1 cells	Inhibited tumor growth	↓*ALDH1A1*; ↓*NANOG*; ↓*CR1*; ↓*GDF3*; ↓*FOXD3*; ↓*NOTCH4*; ↓*WNT3*	50 mg/kg (i.p.)	3, 5 weeks	Castro et al., 2019 [86]
Female athymic BALB/c mice transplanted with KPL-1 cells	Suppressed tumor growth	↑Apoptotic ratio	25, 50 mg/kg (i.p.)	26 days	Kanematsu et al., 2011 [265]
Nude mice xenografted with MDA-MB-453 cells	Reduced tumor size	↑*Egr1*; ↓*cyclin B1*; ↓*CDC25c*	100 mg/kg (i.v.)	15 days	Yang et al., 2016 [106]
*Gastrointestinal tract and associated cancers*					
*Esophageal cancer*					
SCID mice inoculated with BEAC and FLO-1 cells	Reduced tumor size	Not reported	0.75 mg/day (s.c.)	2 weeks	Qazi et al., 2010 [121]
Male BALB/c mice inoculated with ECa109 cells	Decreased tumor size	↑LC3B-II; ↓P62	5 mg/kg, every other day (i.p.)	2 weeks	Lu et al., 2020 [122]
Small intestine					
Male Apc^Min/+^ mice	Decreased tumor number and size	↑Apoptosis; ↓p-JNK; ↓p-ERK; ↓p-Akt	300 and 600 ppm/day (via diet)	3 weeks	Hu et al., 2006 [266]
Male Apc^Min/+^ mice	Reduced tumor size	↑Apoptosis; ↑p21; ↑caspase-3; ↑caspase-9; ↑COX-2; ↓p-Akt	300 and 600 ppm/day (via diet)	3, 10 weeks	Shen et al., 2007 [267]
*Colon cancer*					
Nude male mice xenografted with HCT116 cells	Suppressed tumor growth; decreased tumor size	↑CDK1; ↑MK2; ↑p38 phosphorylation	1 and 5 mg/kg/day (i.p.)	13 days	Byun et al., 2016 [134]
Male C57BL/6J^+/Min^ mice	Inhibited tumor growth	↓HDAC; ↑acetylated histone H4; ↑p21; ↑Bax	~6 µmol/day (via diet)	10 weeks	Myzak et al., 2006 [245]
Male WT and Nrf2 mice induced tumors with DMT	Reduced tumor size	↓HDAC; ↓HDAC3 protein; ↑global histone H4 acetylation	400 ppm/day or alternate days (via diet)	25, 35 weeks	Rajendran et al., 2015 [268]
*Hepatocellular cancer*					
Female BALB/c athymic mice inoculated with HepG2 cells	Reduced tumor growth and volume	Not reported	50 mg/kg, every 2 days (i.p.)	13 days	Wu et al., 2016 [160]
*Pancreatic cancer*					
Male Syrian Hamster injected with BOP to initiate carcinogenesis	Prevented pancreatic carcinogenesis	Not reported	80 ppm/day (p.o.)	3 weeks	Kuroiwa et al., 2006 [269]
Male SCID mice inoculated with PANC-1	Decreased tumor growth	Not reported	250–500 µmol/kg/d (i.p.)	3 weeks	Pham et al., 2004 [163]
Female athymic (nu/nu) mice inoculated with Mia Paca-2	Inhibited tumor growth	Not reported	25 or 50 mg/kg (5× per week i.p.)	4 weeks	Li et al., 2012 [164]
Male NOD/SCID/IL2Rγ mice inoculated with human pancreatic CSCs	Reduced tumor growth	↓Smo; ↓Gli 1; ↓Gli 2; ↓Oct-4; ↓VEGF; ↓PDGFα; ↓Bcl-2; ↓XIAP; ↑E-Cadherin	20 mg/kg/day (5× per week p.o.)	6 weeks	Li et al., 2013 [270]
Nude mice inoculated with MIA-PaCa2	Blocked tumor growth and angiogenesis	↑Apoptosis; ↓NK-κB binding	4.4 mg/kg (i.p.) on days 4, 5, and 6 after tumor transplant	1 week	Kallifatidis et al., 2009 [165]
BALB/c nude mice (transgenic pancreatic cancer mice)	Reduced tumor volume and weight	↑Nrf2; ↓Ki-67; ↑p-AMPK	50 mg/kg, every other day (i.p.)	120 days	Chen et al., 2018 [169]
*Gynecological cancers*					
*Endometrial cancer*					
Female SCID mice inoculated with Ishikawa cells	Reduced tumor volume	↑Apoptosis	50 mg/kg once a day (i.p.)	30 days	Rai et al., 2020 [175]
*Ovarian cancer*					
Athymic mice inoculated with A2780 cells	Inhibited tumor growth	↑IP_3_R	40 mg/kg, once a day (i.p.)	7 days	Hudecova et al., 2016 [180]
*Lung cancer*					
A/J mice treated with benzopyrene and NNK	Inhibited cellular proliferation. Reduced tumor size and weight	↑Apoptosis; ↑casspase-3; ↓PCNA	1.5 and 5 µmol/g (p.o.)	42 weeks	Conaway et al., 2005 [271]
Nude mice inoculated with LTEP-A2 cells	Reduced tumor weight	↑Apoptosis; ↑G2/M arrest	25–100 mg/kg, 3 doses/week (i.p.)	9 days	Liang et al., 2008 [194]
NOD/SCID mice inoculated with A549 cells	Reduced tumor volume and weight	↑Apoptosis; ↑H3 acetylation, ↑H4 acetylation; ↑p53; ↑p21; ↑Bax; ↑G0/G1 arrest; ↑G2/M arrest; ↓HDAC	9 µM/mice/day on alternate days (p.o.)	28 days	Jiang et al., 2016 [197]
BALB/c nu/nu male mice inoculated with NSCLC	Reduced tumor volume	↓EGFR	10 µmol/kg, 5 doses/week (i.t.)	21 days	Chen et al., 2015 [204]
BALB/c nude female inoculated with H1299	Reduced tumor weight and volume and inhibited cell migration and invasion	↑ERK5; ↑pERK5; ↑E-Cadherin; ↑ZO-1; ↓pc-jun; ↓pc-Fos; ↓N-Cadherin; ↓Snail1	25 and 50 mg/kg every 3 days (i.p.)	21 days	Chen et al., 2019 [205]
Nude male BALB/c mice inoculated with H1299 and 95D cells	Decreased the incidence of lung metastasis	↓miRNA-616-5p; ↓β-catenin; ↓N-cadherin; ↓Vimentin	25 or 50 mg/kg, every 3 days (i.v.)	4 weeks	Wang et al., 2017 [200]
*Neurological cancer*					
Female NSG mice inoculated with GBM10 cells	Inhibited tumor growth	Not reported	100 mg/kg for 5-day cycles (p.o.)	3 weeks	Bijangi-Visheshsaraei et al., 2017 [209]
*Skin cancer*					
C57BL/6 mice injected with B16F-10 melanoma cells	Inhibited tumor growth and lung metastasis	↓Lung hydroxyproline; ↓lung uronic acid; ↓lung hexosamine; ↓serum sialic acid; ↓serum GGT; ↑IL-2; ↑IFN-γ; ↓IL-1β; ↓IL-6; ↓TNF-α	500 µg/kg (i.p.)	10 days	Thejass and Kuttan, 2006 [272]; Thejass and Kuttan, 2007 [273]
C57BL/6 mice inoculated with B16 cells	Reduced tumor volume	↓HDAC	500 µmol/kg, 3 doses/week (i.p.)	4 weeks	Do et al., 2010 [218]
C57Bl/6 mice inoculated with B16 cells	Inhibited tumor growth and reduced volume	↓HDAC	500 µmol/kg, 3 doses/week (i.p.)	4 weeks	Enriquez et al., 2013 [217]
NSG mice inoculated with A375	Reduced tumor formation and volume	↑Apoptosis; ↓Ezh2; ↓H3K27me3; ↓MMP-9; ↓MMP-2; ↑TIMP3; ↑PARP cleavage; ↑procaspase-8; ↑procaspase-9	10 µmol/kg, 3 doses/week (p.o.)	6 weeks	Fisher et al., 2016 [215]
*Urogenital cancers*					
*Bladder cancer*					
Nude female athymic mice xenografted with UMUC3 cells	Inhibited tumor growth	Not reported	295 µmol/kg (p.o.)	2 weeks	Abbaoui et al., 2012 [82]
Male athymic mice xenografted with UMUC3 cells	Suppressed tumor growth	↑Apoptosis; ↑caspase-3; ↑cyt. c; ↓survivin	12 mg/kg (p.o.)	5 weeks	Wang and Shan, 2012 [274]
*Prostate cancer*					
Nude male athymic BALB/c (nu/nu) mice xenografted with PC-3 cells	Reduced tumor growth	↑Apoptosis; ↑Bax; ↑Ac-H3; ↑Ac-H4; ↓HDAC	443 mg/kg/day (p.o.)	3 weeks	Myzak et al., 2007 [84]
Male and female athymic mice PC-3 xenograft	Inhibited tumor growth	↑Apoptosis; ↑Bax	5.6 µmol, 3 times/week (p.o.)	3 weeks	Singh et al., 2004 [240]
Male TRAMP [C57BL/6xFVB]F1 hybrid	Decreased cell proliferation and pulmonary metastasis	↑Apoptosis; ↑E-Cadherin; ↑Bad; ↑Bak; ↑Bid; ↑Bax; ↑NK cell cytotoxicity; ↑PARP cleavage; ↓Mcl-1; ↑T-cell infiltration	6 µmol, 3 times/week (p.o.)	17–19 weeks	Singh et al., 2009 [275]
PTEN^ L/L;PB-Cre4 mice	Inhibited cell viability and proliferation	↑Apoptosis; ↑cell cycle arrest; ↑caspase-3; ↑caspase-7; ↑cyclin B1; ↓cyclin D2	0.1, 1 µmol/g/day (p.o.)	8 weeks	Traka et al., 2010 [276]
TRAMP mice	Inhibited tumor growth	↓ACC1; ↓FASN; ↓acetyl-coA; ↓total FFA; ↓phospholipids	Not specified	Not specified	Singh et al., 2018 [262]
TRAMP and Hi-Myc mice with prostate adenocarcinoma	Decreased tumor size	↓Glycolysis; ↓HKII; ↓PKM2; ↓LDHA; ↓lactate	1 mg, 3 times/week (p.o.)	5 weeks	Singh et al., 2019 [253]

Symbols: ↑, increased or upregulated; ↓decreased or downregulated.

**Table 3 cancers-13-04796-t003:** Clinical studies on broccoli constituents in cancer prevention and intervention.

Study Subjects	Study Type	Study Population	No. of Patients/Control Subjects	Intervention	Status	Main Findings/Objectives	References
Healthy males	Dietary intervention study	United Kingdom	20	Brussels sprouts and broccoli (250 g each)	Completed	Increased urinary excretion of PhIP metabolites in group that did not consume broccoli	Walters et al., 2004 [278]
Healthy adults	Randomized placebo-controlled clinical trial	China	200	Broccoli sprout infusion (400 μmol GFN)	Completed	Showed an inverse association of dithiocarbamate and aflatoxin-DNA adducts	Kensler et al., 2005 [279]
Healthy adults	Crossover clinical trail	China	50	Broccoli sprout beverages (800 μmol GFN or 150 μmol SFN)	Completed	Enhanced urinary excretion of mercapturic acids of acrolein, benzene, and crotonaldehyde	Kensler et al., 2012 [280]
Healthy adults	Randomized, placebo-controlled clinical trial	China	148/143	Broccoli sprout beverages (600 μmol GFN and 40 μmol SFN)	Completed	Increased the excretion of benzene-derived mercapturic acid in individuals with positive *GSTT1* genotype compared to null genotype	Egner et al., 2014 [281]
Healthy adults (smokers and non-smokers)	Randomized crossover study	Italy	20	Broccoli (200 g/day)	Completed	Decreased DNA strand break in both smoker and non-smoker and reduced oxidative purine in the smoker group	Riso et al., 2009 [282]
Health adults (smokers)	Placebo-controlled intervention study	Italy	27	Broccoli (250 g/day, 110 μmol ITC/day)	Completed	Decreased oxidized DNA lesions and suppressed DNA strand breaks in the PBMCs of smokers with higher protections with *GSTM1*-null genotype	Riso et al., 2010 [283]
Healthy volunteers	Pilot clinical trial	United State	10	GFN-rich (600 μmol/L GFN) or SFN-rich (150 μmol/L SFN) BSE	Completed	Upregulated *NQO1* mRNA in oral mucosa	Buamen et al., 2016 [284]
Healthy adults	Randomized clinical trial	United States	20	Fresh broccoli sprouts or BSE (200 µmol SFN)	Completed	Decreased PBMC HDAC activity, especially in higher dose group or following repeated intake	Atwell et al., 2015 [79]
Breast cancer patients	Double-blinded clinical trial	United States	27/27	Broccoli seed extract with GFN (224 mg or 512 µmol GFN)	Completed	Decreased PBMC HDAC activity and Ki-67 and HDAC3 serum levels	Atwell et al., 2015 [285]
Breast cancer patients	Double-blinded clinical trial	United States	27/27	Cruciferous vegetables	Completed	Increased consumption led to significantly lower Ki-67 levels in breast cancer samples	Zhang et al., 2015 [286]
Breast cancer patients	Non-randomized interventional clinical trial	Belgium, France, Spain, United Kingdom	60	α-cyclodextrin complex of SFN (SFX-01)	Not completed	Showed potential to reverse resistance to endocrine therapy in metastatic breast cancer	Howell et al., 2019 [287]
Pancreatic cancer patients	Double-blinded, randomized pilot trial	Germany	40	Freeze-dried broccoli sprout (90 mg SFN/day, 507.64 µmol/day)	Not completed	Primary goal is to increase the survival of patients with pancreatic ductal adenocarcinoma	Lozanovski et al., 2014 [288]
Prostate cancer patients	Non-randomized trial	United Kingdom	20	Broccoli (400 g/week)	Completed	Broccoli consumption interacted with *GSTM1* genotype and resulted in alterations of TGFβ1 and EGF signaling pathways	Traka et. al., 2008 [289]
Prostate cancer patients	Single-arm trial	United States	20	SFN-rich extract (200 µmol/day)	Completed	Reduced PSA by more than 50% in one patient and registered a smaller decline (<50%) in PSA in seven patients; prolonged PSA doubling time	Alumkal et al., 2015 [87]
Prostate cancer patients	Double-blinded, randomized, placebo-controlled trial	France	38/40	SFN extracted from broccoli seeds (60 mg SFN/day, 338.42 µmol SFN/day)	Completed	Lowered PSA level at months 0, 1, 3, and 6 as well as prolonged PSA doubling time	Cipolla et al., 2015 [290]
Prostate cancer patients	Randomized double-blinded, controlled trial	United Kingdom	41/20	Broccoli soup (300 mL/week)	Completed	Attenuated the transcriptional changes in the prostate in line with a reduction in the risk of cancer progression	Traka et al., 2019 [291]
At-risk prostate cancer patients	Double-blinded, randomized, controlled trial	United States	48/48	BSE (200 μmol SFN/day)	Completed	Increased HDAC activity in prostate cancer patients; did not alter tissue biomarkers; downregulated *AMACR* and *ARLNC1* genes	Zhang et al., 2020 [90]
Skin cancer patients	Double-blinded, randomized clinical trial	United State	17	BSE (50–200 μmol SFN/day)	Not completed	To evaluate the effect on biomarkers Ki-67, Bcl-2, and STAT3	Kirkwoood et. al., 2016 [292]
Skin cancer patients	Double-blinded, randomized clinical trial	United States	17	BSE (50–200 μmol SFN/day)	Completed	Decreased the levels of pro-inflammatory cytokines (IP-10, MCP-1, MIG, and MIP-1β) and increased tumor suppressor decorin on day 28	Tahata et al., 2018 [89]

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
