# Peer review of "Sulforaphane: A Broccoli Bioactive Phytocompound with Cancer Preventive Potential"

_cancers, 2021, doi:10.3390/cancers13194796_

Round 1

Reviewer 1 Report

This is a well-written review that will be helpful to others studying or planning to study sulforaphane as an anti-cancer compound and in related areas of research.  It states that the literature review was systematic, however it is not clear why endometrial cancer was not covered amoung the gynecologic malignancies discussed.  There was at least one published on sulforphane efficacy against endometrial cancer before April 2021: PMID: 32443471 (PMC7281543 , DOI: 10.3390/cancers12051273 )

and endometrial cancer should be included in this review to assure that it is  thorough. and that endometrial cancer is not overlooked as having potential to be treated by sulforaphane.

Author Response

The authors of this manuscript express their sincere thanks to the reviewer for the critical assessment of this work. The authors have acted upon the recommendations of all reviewers which have resulted in a significant enhancement in the quality of this manuscript. All modifications incorporated in the manuscript are highlighted in red color font. A “point-by-point” response to each and every comment is outlined below.

Comment 1:

This is a well-written review that will be helpful to others studying or planning to study sulforaphane as an anti-cancer compound and in related areas of research. 

Response:

We thank the reviewer for their expertise, time, and effort for reviewing our manuscript. We are deeply encouraged by the generous comments about the quality of our work.

Comment 2:

It states that the literature review was systematic, however it is not clear why endometrial cancer was not covered amoung the gynecologic malignancies discussed.  There was at least one published on sulforphane efficacy against endometrial cancer before April 2021: PMID: 32443471 (PMC7281543, DOI: 10.3390/cancers12051273) and endometrial cancer should be included in this review to assure that it is  thorough. and that endometrial cancer is not overlooked as having potential to be treated by sulforaphane.

Response:

We would like to thank the reviewer for identifying this additional study. Our initial literature search began in January of 2020, so this study had been published after our search had begun but before we had finished in April of 2021. We appreciate you bringing this study to our attention, and it has since been incorporated into our manuscript (page 37, lines 770-780). We have adjusted Figure 2 (page 7) and all in-text citations accordingly, including the tables (page 15 and 29).

Additionally,

  1. The reference list has been modified as we have added a new reference. Special attention is given to conform to the order of references and bibliographic style of the journal.
  2. The entire manuscript has been thoroughly checked and edited to ensure uniform style, organization, and quality.

On behalf of my co-authors, I once again express my sincere thanks to the erudite reviewer for the valuable suggestions and constructive input to improve the quality of our manuscript.

Reviewer 2 Report

This review is very well organized, comprehensive, and accurate in data interpretation and discussion about limitations. The report is also well written and formatted professionally. Easy to follow. Recommend to publish as it is.

Author Response

The authors of this manuscript express their sincere thanks to the reviewer for the critical assessment of this work. The authors have acted upon the recommendations of all reviewers which have resulted in a significant enhancement in the quality of this manuscript. All modifications incorporated in the manuscript are highlighted in red color font. A “point-by-point” response to each and every comment is outlined below.

Comment:

This review is very well organized, comprehensive, and accurate in data interpretation and discussion about limitations. The report is also well written and formatted professionally. Easy to follow. Recommend to publish as it is.

Response:

We would like to thank the reviewer for the kind comment. We have worked to make this manuscript comprehensive and easy to follow for the reader, so we are pleased that this has been recognized by an expert reviewer.

Additionally,

  1. The reference list has been modified as we have added a new reference. Special attention is given to conform to the order of references and bibliographic style of the journal.
  2. The entire manuscript has been thoroughly checked and edited to ensure uniform style, organization, and quality.

On behalf of my co-authors, I once again express my sincere thanks to the erudite reviewer for the valuable suggestions and constructive input to improve the quality of our manuscript.

Reviewer 3 Report

 In this review, authors performed an extensive literature search on the effects of SFN in several biological models, as well as clinical studies, to evaluate its potential as a chemopreventative molecule. Extensive tables with a summary of the studies are helpful. However, while there is an extensive description of the studies, the paper reads more like a list of studies and less a discussion and integration of the several reports. As is, it is of little interest to a wider audience. I suggest, in order to not add more length to the manuscript, to add several figures that would summarize the findings, providing an analysis from different perspectives. For instance, for the several effects such as apoptosis and proliferation, add a pathway figure and note the effects visually, for instance also marking with the reference number (of the study or studies showing that effect). Another important issue is a comparison between the different types of cancers and models, what effects are for the most part consistent, and which may be specific to each cancer? This type of analysis and integration of literature reports are the most important part of reviews. As it stands, only one pathway has been detailed in a figure and it is not clear why it has been singled out… By adding the figures with the pathways in the different processes it also allows the reader to understand the cellular effect of the different molecular alterations discussed. I suggest adding the figures in the main text and having the tables as supplementary material. In its current form, the paper will only be of interest to experts in a narrow field, and I believe it would be much more impactful if it would appeal to a wider audience.

Author Response

The authors of this manuscript express their sincere thanks to the reviewer for the critical assessment of this work. The authors have acted upon the recommendations of all reviewers which have resulted in a significant enhancement in the quality of this manuscript. All modifications incorporated in the manuscript are highlighted in red color font. A “point-by-point” response to each and every comment is outlined below.

Comment 1:

In this review, authors performed an extensive literature search on the effects of SFN in several biological models, as well as clinical studies, to evaluate its potential as a chemopreventative molecule. Extensive tables with a summary of the studies are helpful. However, while there is an extensive description of the studies, the paper reads more like a list of studies and less a discussion and integration of the several reports. As is, it is of little interest to a wider audience.

Response:

We would like to thank this reviewer for their insight and critical analysis of our manuscript. While we agree that the main body of our manuscript consists of an extensive description of the studies, we have organized it in a way that is easy to follow for the reader. Within each subsection, the studies have been grouped together based on the mechanistic effect of SFN allowing for an integrative reading experience. Because of the way the main body of our manuscript is written/organized, we believe that it can easily reach a broad audience as well as appeal to colleagues pursuing research in this field. Additionally, the tables were created with the intention of providing the reader with a tool that can aid in discussion and act as a quick reference, therefore facilitating the ease of the reading experience. 

Comment 2:

I suggest, in order to not add more length to the manuscript, to add several figures that would summarize the findings, providing an analysis from different perspectives. For instance, for the several effects such as apoptosis and proliferation, add a pathway figure and note the effects visually, for instance also marking with the reference number (of the study or studies showing that effect).

Response:

We would like to thank the reviewer for the suggestion of incorporating an additional figure depicting the apoptosis pathway. Because this was one of the most commonly targeted anticancer mechanisms identified in our research, we agree that it would be beneficial to create a visual representation for the readers. Accordingly, we have added a new figure (Figure 5, page 53) and have referenced it in the appropriate paragraph of our discussion (page 52, lines 1412 and 1413). It is our hope that this additional figure will provide a more integrative experience for the readers.

Comment 3:

Another important issue is a comparison between the different types of cancers and models, what effects are for the most part consistent, and which may be specific to each cancer? This type of analysis and integration of literature reports are the most important part of reviews.

Response:

The “Conclusion and Future Directions” section of our manuscript elaborates on the common mechanisms observed in the studies reviewed (page 52, lines 1400-1417). This section describes the Nrf2/Keap pathway, which was observed to be modified by SFN in many of the included in vitro and in vivo studies. This section also discusses induction of apoptosis and cell cycle arrest that was noted in nearly every study. Additionally, Figure 3 (page 52) visually represents the wide range, but commonly observed, effects that SFN had on many cell types. Finally, unique effects of SFN on different cancer types have been thoroughly described under the heading of each cancer type.

Comment 4:

As it stands, only one pathway has been detailed in a figure and it is not clear why it has been singled out… By adding the figures with the pathways in the different processes it also allows the reader to understand the cellular effect of the different molecular alterations discussed. I suggest adding the figures in the main text and having the tables as supplementary material.

Response:

While we agree that an additional figure depicting the apoptosis mechanism would be beneficial to the reader, we intend to keep our tables in the main text. We feel as though these tables are fundamental aspects of our manuscript and are important distinguishing features of our comprehensive review of current literature. To add them as supplemental materials would compromise the purpose as well as the quality of our manuscript.

Comment 5:

In its current form, the paper will only be of interest to experts in a narrow field, and I believe it would be much more impactful if it would appeal to a wider audience.

Response:

We appreciate learning the review’s opinion. With the revisions that we have made in response to the reviewer’s comments, we hope that the scope of our paper can be appreciated as it has been supported by positive evaluations from three additional reviewers. While this manuscript details specifics of the anticancer properties of SFN, it is our sincere hope that our comprehensive review would provide a valuable resource to readers and galvanize future research on a remarkable dietary phytochemical SFN.

Additionally,

  1. The reference list has been modified as we have added a new reference. Special attention is given to conform to the order of references and bibliographic style of the journal.
  2. The entire manuscript has been thoroughly checked and edited to ensure uniform style, organization, and quality.

On behalf of my co-authors, I once again express my sincere thanks to the erudite reviewer for the valuable suggestions and constructive input to improve the quality of our manuscript.

Reviewer 4 Report

The manuscript is impressive, well written and interesting.  It includes a complete overview of the subject that will contribute to the literature.

It is well structured and described cohesively.

As it is organized by cancer type, it adds a new perspective and distinguishes this review from other ones, that were published in recent years.

Author Response

The authors of this manuscript express their sincere thanks to the reviewer for the critical assessment of this work. The authors have acted upon the recommendations of all reviewers which have resulted in a significant enhancement in the quality of this manuscript. All modifications incorporated in the manuscript are highlighted in red color font. A “point-by-point” response to each and every comment is outlined below.

Comment:

The manuscript is impressive, well written and interesting.  It includes a complete overview of the subject that will contribute to the literature.

It is well structured and described cohesively.

As it is organized by cancer type, it adds a new perspective and distinguishes this review from other ones, that were published in recent years.

Response:

We would like to thank the reviewer for these encouraging comments. While this manuscript may be a bit long because you have tried our best to include an impressive body of knowledge, we have worked to make it cohesive and well-organized. We are pleased that these aspects of our manuscript have been appreciated by an expert reviewer.

Additionally,

  1. The reference list has been modified as we have added a new reference. Special attention is given to conform to the order of references and bibliographic style of the journal.
  2. The entire manuscript has been thoroughly checked and edited to ensure uniform style, organization, and quality.

On behalf of my co-authors, I once again express my sincere thanks to the erudite reviewer for the valuable suggestions and constructive input to improve the quality of our manuscript.